# REWARDRANK: OPTIMIZING TRUE LEARNING-TO-RANK UTILITY

## ABSTRACT

Traditional ranking systems optimize offline proxy objectives that rely on oversimplified assumptions about user behavior, often neglecting factors such as position bias and item diversity. Consequently, these models fail to improve true counterfactual utilities such as such as click-through rate or purchase probability, when evaluated in online A/B tests. We introduce RewardRank, a data-driven learning-to-rank (LTR) framework for counterfactual utility maximization. RewardRank first learns a reward model that predicts the utility of any ranking directly from logged user interactions, and then trains a ranker to maximize this reward using a differentiable soft permutation operator. To enable rigorous and reproducible evaluation, we further propose two benchmark suites: (i) Parametric Oracle Evaluation (PO-Eval), which employs an open-source click model as a counterfactual oracle on the Baidu-ULTR dataset, and (ii) LLM-as-User Evaluation (LAU-Eval), which simulates realistic user behavior via large language models on the Amazon-KDD-Cup dataset. RewardRank achieves the highest counterfactual utility across both benchmarks and demonstrates that optimizing classical metrics such as NDCG is sub-optimal for maximizing true user utility. Finally, using real user feedback from the Baidu-ULTR dataset, RewardRank establishes a new state of the art in offline relevance performance. Overall, our results show that learning-to-rank can be reformulated as direct optimization of counterfactual utility, achieved in a purely data-driven manner without relying on explicit modeling assumptions such as position bias.

## 1 INTRODUCTION

The goal of any ranking system is to model human decision-making in a way that maximizes user engagement and utility. However, real-world user behavior is shaped by subtle, context-dependent cognitive biases that traditional ranking losses fail to capture. Engagement often drops when users are presented with redundant or overly similar items, whereas introducing diversity or strategically positioning items can significantly enhance interest. For example, the decoy effect—where the presence of a less-attractive item increases preference for a similar alternative—has been observed in search interactions and shown to meaningfully influence user choices (Wang et al., 2025a). Other well-documented biases include position bias (Chen et al., 2024; Hager et al., 2024; Zou et al., 2022), brand bias (Li et al., 2025), and similarity aversion (Tversky & Simonson, 2004). In online advertising, the goal is often to maximize the probability that a user clicks on any item in the list, rather than just the top-ranked one. If data shows that users tend to click on the second position, it may be optimal to place the most engaging ad there to improve overall performance. Likewise, in recommendation scenarios, users may prefer a diverse mix of product styles or brands over a cluster of nearly identical, albeit highly relevant, items. Traditional ranking losses, which emphasize relevance at individual positions (typically the top), are ill-suited for modeling such list-level behaviors (Figure 1). They overlook the fact that user utility depends not just on which items are shown, but how they are arranged, highlighting the limitations of handcrafted objectives in capturing the interactive and comparative nature of real user decision-making.

A natural way to model user behavior is by learning preferences over full permutations of items within a query group (i.e., a query and its associated items). The ideal objective is to identify and rank those permutations that are most likely to drive user engagement, which can be formulated as a likelihood maximization problem: maximizing the probability of observing high-

engagement permutations while minimizing that of unengaged ones. However, the combinatorial explosion of the permutation space quickly renders this approach intractable; for instance, ranking 10 items results in 10! (over 3.6 million) possible arrangements. To address this, recent approaches adopt a utility-based framework (Feng et al., 2021; Shi et al., 2023; Xi et al., 2024; Ren et al., 2024; Wang et al., 2025b), where a utility model is trained to score permutations based on user preferences, and a ranker is subsequently optimized to generate item orders that maximize the predicted utility. While this framework reduces the combinatorial burden, it introduces two key challenges. First is the classic exploration–exploitation dilemma: the ranker must leverage known high-utility arrangements while also exploring novel permutations that may yield higher engagement.

Second is utility model misspecification (akin to reward misspecification Clark & Amodei (2016); Coste et al. (2023)): if the learned reward model fails to accurately reflect true user preferences, the ranker may be misled, resulting in poor exploration and degraded overall performance.

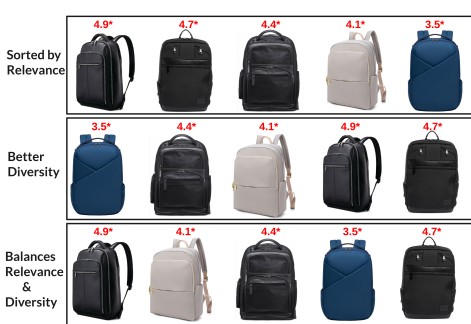

In operational ranking systems, user interactions are logged for only a small fraction of the total permutation space. For example, in Figure 1, only 3 out of the 120 possible arrangements of 5 items for the query "laptop bag" are observed, constituting the factual space. These observed interactions define the *factual/observed* space, whereas the vast majority of unexposed, yet potentially high-utility, permutations form the *counterfactual/unobserved* space. The optimal arrangement that maximizes user engagement may exist anywhere within the full permutation space of 120 arrangements. One of the major challenges in counterfactual ranking lies in reliably evaluating unobserved permutations. Even if the full permutation space is modeled, evaluating ranking strategies under counterfactual settings remains challenging due to the lack of explicit supervision (Agarwal et al., 2019; Gupta et al., 2024b;a; Buchholz et al., 2024). For instance, in Figure 1, 117 out of 120 possible arrangements remain unobserved, making their evaluation inherently counterfactual. Existing approaches, such as offline A/B testing, inverse propensity scoring, or other debiasing techniques, are often costly, statistically unstable, or difficult to scale, making counterfactual evaluation a central bottleneck in listwise utility optimization.

Figure 1: **Counterfactual ranking with true learning-to-rank utility**. Three arrangements for the query "*laptop bag*", with item relevance/rating scores (0–5*). The top row ranks purely by relevance but suffers from similarity aversion due to identical color and style, lowering engagement. The middle row improves diversity but surfaces low-relevance items early, which may deter clicks. The bottom row balances diversity and relevance, placing distinct yet relevant items in top positions, leading to higher predicted utility and user engagement. (Figures are generated by GPT-4o)

To address these challenges, we propose REWARDRANK, a counterfactual utility maximization framework that models user behavior over full item permutations. Rather than scoring items in isolation, we learn a permutation-aware utility function that captures user preferences at the list level. To enable differentiable optimization over permutations, we employ the *SoftSort* operator (Prillo & Eisenschlos, 2020) to construct soft item embeddings, allowing end-to-end training of the ranking model with respect to utility gradients. To mitigate the effects of reward model misspecification—where the learned utility may diverge from actual user preferences—we introduce a correction term in the ranker's training objective that improves robustness during optimization. For evaluation, we present two scalable, fully automated protocols that assess counterfactual performance without requiring human labels. *Parametric Oracle Evaluation (PO-Eval)* uses a pretrained, position-aware oracle to provide soft supervision and serve as a proxy for user behavior. *LLM-As-User Evaluation (LAU-Eval)* leverages large language models to simulate user preferences and assess ranking quality across unobserved permutations. Together, these methods enable efficient benchmarking of counterfactual ranking strategies and help align learned rankings with actual or simulated user utility.

Our key contributions can be summarized as:

- We introduce REWARDRANK, a framework for counterfactual utility maximization that learns a permutation-aware reward model, capturing human list-level preferences and behavioral biases without any explicit modeling assumptions such as position bias.

- We enable end-to-end ranking optimization using differentiable soft permutation operators, and incorporating a per-item auxiliary loss along with a misspecified reward correction term to aid counterfactual space exploration.

- We propose two large-scale automated evaluation protocols: *PO-Eval* (parametric oracle) and *LAU-Eval* (LLM-as-user), and construct reproducible testbeds for scalable counterfactual ranking evaluation. Experiments on these testbeds reveal that optimizing standard offline ranking metrics such as NDCG do not reliably maximize true user utility.

- In both proposed counterfactual testbeds, REWARDRANK consistently achieves the highest learning-to-rank (LTR) utility compared to existing and widely adopted ranking methods. When trained on real click signals from an industry-scale dataset, REWARDRANK further establishes a new state-of-the-art in relevance performance.

## 2 RELATED WORK

**Traditional ranking methods.** Traditional learning-to-rank (LTR) methods are typically categorized into three classes: point-wise, pair-wise, and list-wise approaches. Point-wise methods treat ranking as a regression or classification problem by independently assigning relevance scores to each item (Burges et al., 2005a;b). While computationally efficient, they neglect interactions among items in the ranked list. Pair-wise approaches, including RankSVM (Joachims, 2002), RankBoost (Freund et al., 2003), and LambdaMART (Burges, 2006; Wu et al., 2010), aim to learn relative preferences between item pairs, improving over point-wise methods but still failing to capture full list-level dependencies. In contrast, list-wise methods optimize objectives over the entire ranking, such as NDCG (Cao et al., 2007; Xia et al., 2008), offering better alignment with evaluation metrics.

Recent large-scale datasets such as Baidu-ULTR (Zou et al., 2022) have enabled realistic benchmarking of ranking algorithms under user-interaction-driven settings, facilitating systematic studies on position bias, distribution shift, and counterfactual evaluation in LTR (Hager et al., 2024). Building on these advances, modern approaches have expanded beyond purely supervised objectives toward *data-driven* and *representation-rich* formulations. Pretraining-based LTR models leverage large language or multimodal corpora to learn transferable ranking priors (Hou et al., 2024), while latent cross-encoding methods (Luo et al., 2022) and set-aware transformers Qin et al. (2021) jointly embed queries and items to capture fine-grained contextual dependencies.

**Counterfactual Learning-to-Rank.** Prior work in counterfactual learning-to-rank (CLTR) primarily addresses position bias in implicit feedback using methods such as inverse propensity scoring (IPS) (Joachims et al., 2017) and doubly robust estimation (Oosterhuis, 2023). Extensions include modeling trust bias (Agarwal et al., 2019) and jointly correcting for both position and trust biases (Vardasbi et al., 2020). Recent approaches explore policy optimization via proximal updates (Gupta et al., 2024b) and extend this to trust-aware CLTR through proximal ranking objectives (Gupta et al., 2024a). While effective, these methods often focus narrowly on position bias or make strong assumptions, underscoring the need for broader utility-driven ranking frameworks, as pursued in this work.

**Utility-oriented counterfactual reranking**. Reranking methods enhance an initial ranked list by applying a secondary model to better optimize downstream objectives such as user utility, fairness, or diversity (Xi et al., 2024; Wang et al., 2025b). Recent work in counterfactual ranking predominantly follows a two-stage framework consisting of a generator and an evaluator (Xi et al., 2024; Shi et al., 2023; Ren et al., 2024; Wang et al., 2025b). For example, URCC (Xi et al., 2024) trains a set-aware utility model and employs a context-sensitive pairwise LambdaLoss to guide the ranker. NLGR (Wang et al., 2025b) leverages neighboring lists within a generator-evaluator setup for utility optimization. PRS (Feng et al., 2021) adopts beam search to generate candidate permutations and evaluates them using a permutation-wise scoring model, while PIER (Shi et al., 2023) uses SimHash to select top-K candidates from the full permutation space efficiently.

Reranking approaches rely on a strong base ranker trained on logged data and typically explore counterfactuals around its initial permutations Xi et al. (2024); Wang et al. (2025b), which constrains exploration and limits discovery of globally optimal rankings. Importantly, these methods do not learn

an explicit reward model; instead, they assume a predefined metric such as NDCG to serve as the counterfactual reward (Joachims et al., 2017; Agarwal et al., 2019). While effective in certain settings, this reliance on a fixed evaluation metric can hinder adaptability to more general or task-specific reward signals.

**Differential approximation to ranking**. A key challenge in learning-to-rank is the mismatch between evaluation metrics (e.g., NDCG, MAP) and surrogate loss functions amenable to gradient-based optimization, due to the non-differentiable nature of sorting operations. To address this, prior work has either proposed smooth approximations to the rank function (e.g., ApproxNDCG (Qin et al., 2010)) or introduced differentiable approximations to argsort using soft permutation matrices (Grover et al., 2019; Prillo & Eisenschlos, 2020); for instance, PiRank (Swezey et al., 2021) and NeuralNDCG (Pobrotyn & Białobrzeski, 2021) utilize NeuralSort as a temperature-controlled surrogate. Another line of work leverages the Plackett–Luce distribution to model ranking policies in a differentiable manner (Oosterhuis, 2021). Methods like PG-RANK (Gao et al., 2023) use policy gradients to optimize the expected reward over the Plackett–Luce distribution based on REINFORCE, while ListNet (Cao et al., 2007) and ListMLE (Xia et al., 2008) employ the Plackett–Luce framework to derive smooth list-wise objectives.

Please refer to Appendix Section A.5 for additional related work.

## 3 Learning-To-Rank Problem: Utility Maximization vs Sorting

A data sample of an LTR problem is a *query group* (QG) consisting of a query, $q$, and a set of $L$ items, $\{x_\ell\}_{\ell=1}^L$, where $L$ may vary. The *query* may represent, for example, a search string, a user profile, or other contextual information like device type and page layout. The *items* are candidate entities like webpages, songs, or products retrieved by an upstream system. We assume that the QGs are drawn i.i.d. from a distribution $\mathcal{P}$, i.e. $(q, \{x_\ell\}) \sim \mathcal{P}$. When a user is presented with a ranking/arrangement (permutation), $\pi : [L] \to [L]$, of the items of a QG, i.e. $(x_{\pi(1)}, \ldots, x_{\pi(L)})$, they interact with the ranked items, yielding a stochastic utility $U(q, \{x_\ell\}, \pi) \in \mathbb{R}$, which is a hidden function of the QG and the ranking. In typical internet systems, the utility can represent outcomes such as whether a user clicks or purchases any item, or continuous measures such as the total minutes of media consumed. Our objective is to learn a ranking policy, $f$, mapping the QGs to permutations, that maximizes the expected utility return, i.e.

$$f^* = \underset{f}{\arg\max} \, \mathbb{E}_{(q, \{x_\ell\}) \sim \mathcal{P}}[U(q, \{x_\ell\}, \pi = f(q, \{x_\ell\}))] \tag{1}$$

Based on the choice of the utility, this objective corresponds to business metrics like click-through rate, units sold, or streamed minutes. The main challenge here is that the hidden stochastic utility function $U$ is not directly observable. Instead we are given a training dataset, $\mathcal{D}$, consisting of $N$ QGs (indexed by $i$) and their observed utility $\{u_i\}$ under some logged rankings $\{\pi_i\}$, i.e. $\mathcal{D} = \{(q_i, \{x_{i,\ell}\}_{\ell=1}^{L_i}, \pi_i, u_i)\}_{i \in [N]}$. We assume that similar hold-out test and validation dataset are also available. This setting can be viewed as an offline one-step reinforcement-learning problem in which the state space is comprised of all possible QGs in the support of $\mathcal{P}$, the action space is comprised of all item permutations, and the reward is the observed utility.

In practice, most QGs are unique, so we observe only one out of $L!$ possible rankings for each. Consequently, even if we propose a better alternative ranking for a given QG, the *counterfactual* utility it would have obtained remains unknown. To address this, traditional LTR algorithms optimize heuristic offline ranking metrics like Normalized Discounted Cumulative Gain (NDCG) (Järvelin & Kekäläinen, 2002; Burges, 2006), averaged over a test set. When a user interacts with a ranked QG, we also obtain per-item feedback signals $\{y_\ell \geq 0\}$ (e.g. whether an item was clicked or purchase, or how many minutes it was streamed). Usually, the overall QG-level utility $u$ is some function of these per-item signals. Then, the NDCG of any new ranking $\widehat{\pi}$, on a QG with feedbacks $\{y_\ell\}$ can be defined as

$$\mathrm{NDCG}(\widehat{\pi}, \{y_\ell\}) \triangleq \frac{\mathrm{DCG}(\widehat{\pi}, \{y_\ell\})}{\mathrm{DCG}(\pi^*, \{y_\ell\})} \in [0, 1] \,, \text{ where } \mathrm{DCG}(r, \{y_\ell\}) \triangleq \sum_{\ell=1}^L \frac{2^{y_\ell} - 1}{\log_2(1 + r^{-1}(\ell))} \,. \tag{2}$$

DCG assigns a gain $2^{y_\ell} - 1$ for the item $x_\ell$ in a test QG, but its contribution to the metric is discounted by its position $r^{-1}(\ell)$ under the ranking $r$. Thus, NDCG is maximized when the items are ranked

in the descending order of their feedback values, i.e., under the optimal ranking $\pi^*$. Traditional LTR methods (Burges, 2006; Swezey et al., 2021), aim to maximize NDCG by optimizing various continuous relaxations of it. This heuristic of learning to move items with higher feedback signal to the top of list have been highly successful, potentially because (i) items with positive feedback are usually relevant, and (ii) users tend to focus their attention on the top of the list. However, such offline metrics are now well-known to be sub-optimal as they do not perfectly align with the true (hidden) utility (1) we aim to maximize (Wang et al., 2023; Jeunen et al., 2024). A key advantage of REWARDRANK over traditional LTR methods is its ability to *leverage data without click/purchase labels*. Whereas standard pipelines often discard sessions with no purchases/clicks (or treat them as uninformative negatives), our approach can still extract signal from these interactions via its utility modeling and preference estimation. This aligns with recent evidence that leveraging unlabeled or weakly labeled interaction data—e.g., through pretraining or preference modeling—improves ranking quality (Hou et al., 2024). In the next section, we introduce REWARDRANK, a data-driven ranking framework that directly maximizes the true LTR utility without relying on heuristics or specific user-behavior assumptions.

## 4 REWARDRANK: DATA-DRIVEN LTR UTILITY MAXIMIZATION

In this section, we present the REWARDRANK framework, which aims to maximize the true (hidden) LTR utility defined in (2). At a high level, REWARDRANK proceeds in two stages. First, using the logged training data $\mathcal{D}$, it learns a reward model that predicts the counterfactual utility for any QG and permutation. Then it trains a ranker using the reward model's predictions as supervision, so as to maximize the expected counterfactual LTR utility of the ranker's item arrangement (ranking) policy.

### 4.1 STAGE 1: LEARNING THE UTILITY USING A REWARD MODEL

Let $g(q, \{x_\ell\}, \pi; \phi)$ denote the reward model (parameterized with $\phi$) to predict the scalar utility for the QG $(q, \{x_\ell\})$ and a ranking $\pi$. It is trained solely on the logged query groups, rankings and observed utilities in the training dataset $\mathcal{D}$. When the utility $U \in \{0, 1\}$ is a binary random variable (e.g. click, purchase), we train $g \in [0, 1]$ by minimizing the average binary cross-entropy loss between the observed $u_i$ and the predicted utilities $\widehat{u}_i(\phi) := g(q_i, \{x_{i\ell}\}, \pi_i; \phi)$ over all $i \in [N]$:

$$\min_\phi \left[ \text{RewardLoss}(\phi) \triangleq -\frac{1}{N} \sum_{i=1}^N [u_i \log(\widehat{u}(\phi)) + (1 - u_i) \log(1 - \widehat{u}(\phi))] \right]. \tag{3}$$

When the true utility is a continuous random variable (e.g. minutes a song is streamed) we can use regression losses such mean squared error (MSE) $\min_\phi (1/N) \sum_{i=1}^N \|u_i - \widehat{u}_i(\phi)\|^2$. In our experiments, the reward model is instantiated with a transformer encoder, `Enc`, due to its ability to model functions over sequences (ranked list of items). Before passing a QG into `Enc` each query-item pair $[q, x_\ell]$ is embedded using a text encoder to create the token embedding $\mathbf{e}_\ell$. Next, the ranking of the items $\pi$ is encoded through position encodings $\{\mathbf{p}_k\}$. Then the position encoded tokens are passed to the transformer `Enc`. Finally, the predicted utility is computed as the sigmoid of a linear function of the [CLS] token output. This can be succinctly represented as:

$$g(q, \{x_\ell\}_{\ell=1}^L, \pi; \phi) = \sigma \left[ \mathbf{v}^\top \text{Enc}_{\text{reward}}^{\text{[CLS]}} \left( \{\mathbf{e}_{\pi(k)} + \mathbf{p}_k\}_{k=1}^L \right) \right], \tag{4}$$

where $\mathbf{e}_{\pi(k)}$ is the token embedding of the query and the $k$-th ranked item. During the second stage of training the ranker, we freeze the reward model parameters $\phi$.

**Auxiliary per-item predictor**: Typically the observed utility $u$ is a byproduct of user's interaction with the items. So, we hypothesize that predicting the per-item feedback signals $\{y_\ell\}$ as an auxiliary task would improve the overall quality of the LTR utility prediction. Thus we include an auxiliary prediction head on the item tokens' outputs to predict the feedback signal observed at each ranked position $k \in [L]$. When $y_\ell \in \{0, 1\}$ is binary, the predictions can be instantiated as

$$\widehat{y}_k(\phi) \triangleq \sigma \left[ \widetilde{\mathbf{v}}^\top \text{Enc}_{\text{reward}}^{(k)} \left( \{\mathbf{e}_{\pi(k)} + \mathbf{p}_k\}_{k=1}^L \right) \right], \quad \forall k \in [L], \tag{5}$$

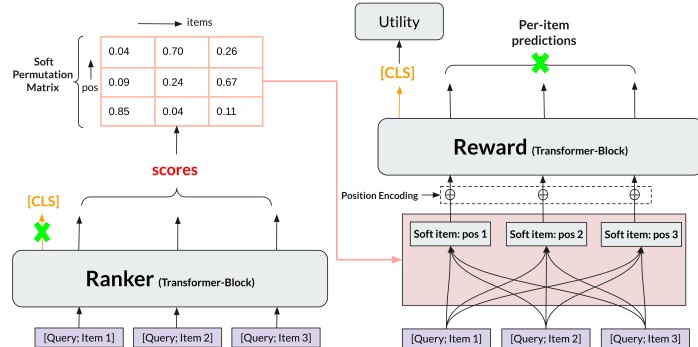

Figure 2: **REWARDRANK.** A ranker scores the items in a query group. These scores are used to compute soft item embeddings via a soft permutation matrix. Position encoded soft item embeddings are passed into a reward to estimate its utility. Finally, the ranker is optimized to maximize the predicted utility.

where $\text{Enc}_{\text{reward}}^{(k)}$ is the output token at the $k$-th position and $\sigma$ is the sigmoid function. We can learn $\widehat{y}_k(\phi)$ alongside $\widehat{u}(\phi)$ by adding the average cross-entropy loss between $\widehat{y}_k$ and $y_{\pi(k)}$,

$$\text{ItemLoss}(\phi) \triangleq -\frac{1}{\sum_i L_i} \sum_{i=1}^{N} \sum_{k=1}^{L_i} [y_{i\pi(k)} \log(\widehat{y}_{ik}(\phi)) + (1 - y_{i\pi(k)}) \log(1 - \widehat{y}_{ik}(\phi))]. \tag{6}$$

as an additional regularizer to $\text{RewardLoss}$ (3). Note that during the training of the ranker in the next stage, these auxiliary predictions can be discarded. Our ablation in Section 5.1 shows that the per-item loss provides a moderate boost in performance. We also apply the per-item loss to query groups (QGs) with no purchases (i.e., no positive labels). This enables us to exploit otherwise discarded sessions and stabilize learning in sparse-feedback regimes by providing item-level signals even when list-level purchase supervision is absent.

## 4.2 STAGE 2: RANKER REWARD MAXIMIZATION THROUGH SOFT SORTING

Typically, rankers are modeled as scoring functions that assign a score to each item in a QG. Then the items are ordered in the descending order of their scores to obtain the final ranking. We follow the same pattern and define $f(q, \{x_\ell\}; \theta)$ as a scoring-based ranker which maps a QG $(q, \{x_\ell\})$ to a set of item scores $\{s_\ell\}$. Following our reward model design, we instantiate $f$ using the same transformer backbone architecture. Since QG has an unordered set of items, we do not use position encoding. Finally the score is computed as the linear function of the output item tokens, i.e.

$$s_\ell \triangleq f_\ell(q, \{x_\ell\}; \theta) \triangleq \mathbf{w}^\top \text{Enc}_{\text{ranker}}^{(\ell)} \left( \{\mathbf{e}_\ell\}_{\ell=1}^L \right), \quad \forall \ell \in [L], \tag{7}$$

where $\text{Enc}_{\text{ranker}}^{(\ell)}$ is the output token of the $\ell$-th item. Our goal is to optimize the effective ranking $\widehat{\pi}$ induced by these scores so that it maximizes the expected counterfactual utility, which is a hidden from us. This is where the reward model comes in handy, as it helps us predict the counterfactual utility as $\widehat{u} := g(q, \{x_\ell\}, \widehat{\pi})$. However, since sorting (of the scores) is a discontinuous operation, it is challenging to optimize the scores to maximize the reward. To enable an end-to-end optimization of the scorer $f$, we resort to a continuous relaxation of the sorting operation.

**Soft Permutation via SoftSort.** *SoftSort* (Prillo & Eisenschlos, 2020) is a continuous relaxation of sorting operation. It defines a *unimodal row-stochastic* matrix (Swezey et al., 2021) as the *soft* permutation matrix $\widehat{\Pi}^{(\tau)} \in [0, 1]^{L \times L}$. Row $k$ of this matrix corresponds to a probability distribution of the $k$-the ranked item over the set of all items. Formally, we define

$$\widehat{\Pi}_{k,\ell}^{(\tau)} \triangleq \frac{\exp\left(-\frac{1}{\tau} \left|s_\ell - s_{\widehat{\pi}(k)}\right|\right)}{\sum_{\ell'=1}^{L} \exp\left(-\frac{1}{\tau} \left|s_{\ell'} - s_{\widehat{\pi}(k)}\right|\right)}, \quad \forall k, \ell \in [L], \tag{8}$$

where $\tau$ is a temperature parameter and $\widehat{\pi}(k)$ is the $k$-th ranked items when (hard) sorting by the scores $\{s_\ell\}$. $\widehat{\Pi}^{(\tau)}$ is a continuous function of the scores $\{s_\ell\}$ and when $\tau \to 0$, $\widehat{\Pi}^{(\tau)}$ tends to the binary hard-permutation matrix $\widehat{\Pi}$, where

$$\lim_{\tau \to 0} \widehat{\Pi}_{k,\ell}^{(\tau)} = \widehat{\Pi}_{k,\ell} \triangleq \mathbb{I}\{\widehat{\pi}(k) = \ell\}, \quad \forall k, \ell \in [L], \tag{9}$$

assuming the scores are unique. Using this soft permutation matrix, we can compute a soft item embedding $\widehat{e}_k^{(\tau)}$ at position $k$ as the following convex combination of the true item embeddings

$$\widehat{\mathbf{e}}_k^{(\tau)} \triangleq \sum_{\ell \in [L]} \widehat{\Pi}_{k,\ell}^{(\tau)} \mathbf{e}_\ell. \tag{10}$$

It is easy to verify that $\widehat{\mathbf{e}}_k^{(\tau)} \to \mathbf{e}_{\widehat{\pi}(k)}$ when $\tau \to 0$. Note that there are alternate soft permutation matrices like NeuralSort (Grover et al., 2019), but we adopt SoftSort for its simplicity and state of the art performance (Prillo & Eisenschlos, 2020). We then compute a soft reward for these soft item embeddings using

$$\widehat{g}(\theta) \triangleq g(q, \{x_\ell\}, \widehat{\Pi}^{(\tau)}) \triangleq \sigma\left[\mathbf{v}^\top \mathtt{Enc}_{\mathrm{reward}}^{\mathtt{[CLS]}}\left(\left\{\widehat{\mathbf{e}}_k^{(\tau)} + \mathbf{p}_k\right\}_{k=1}^L\right)\right], \tag{11}$$

This allows us to compute an approximate predicted reward (11) as a continuous function over the ranker scores $\{s_\ell\}$ through the SoftSort matrix. Finally, we optimize the parameters of scorer $f$ to maximize the average approximate reward over the training set in an end-to-end manner:

$$\min_\theta \left[\mathrm{RankerLoss}(\theta) \triangleq -\frac{1}{N} \sum_{i=1}^N \widehat{g}_i(\theta)\right]. \tag{12}$$

Even though REWARDRANK is maximizing the predicted utility of the soft ranking, we hypothesize that it generalizes well and produces rankings with higher expected counterfactual utility than prior LTR methods.

An alternative to *soft-permutation matrices* is the Plackett–Luce (PL) model, which offers efficient, closed-form gradients for ranking. However, counterfactual learning with PL requires Monte Carlo sampling, leading to high-variance estimates in large action spaces. While variance reduction helps (Gao et al., 2023), unbiased learning fundamentally depends on stochastic logging, which is incompatible with real-world deterministic rankers designed for stability and trust. Soft permutation relaxations like SoftSort (Prillo & Eisenschlos, 2020) approximate permutations in continuous space, enabling gradient-based optimization without sampling. Though computationally more expensive, they reduce variance and support end-to-end utility maximization. We pair SoftSort with a learned reward model that generalizes over logged data, enabling scalable training under deterministic logs. This approach trades unbiasedness for stability and practicality in real-world ranking systems.

**Mitigating reward misspecification.** One challenge of reward modeling the hidden counterfactual utility is model misspecification, i.e. a gap between the predicted and the true utilities. A misspecified reward can misguide the ranker into wrong ranking policies Coste et al. (2023); Clark & Amodei (2016). To mitigate this issue we propose a sample reweighting scheme which modifies the ranker loss as

$$\mathrm{RankerLoss}^{(\lambda)}(\theta) \triangleq -\frac{1}{N} \sum_{i=1}^N w_i \cdot \widehat{g}_i(\theta), \text{ where } w_i = 1 - \lambda|u_i - \widehat{u}_i| \in [0, 1] \text{ and } \lambda \in [0, 1], \quad \forall i. \tag{13}$$

Above loss is a pessimistic upperbound to $\mathrm{RankerLoss}(\theta)$ (12). This reward down-weighting scheme is motivated by a conjecture that when the observed utility $u_i$ for the $i$-th training QG and the corresponding prediction $\widehat{u}_i(\theta)$ are different, the utility prediction on new ranking of this QG would also be less reliable. Through an ablation in Section 5.1 we show that reward misspecification correction slightly improves the REWARDRANK performance.

## 5 EXPERIMENTAL RESULTS

**Datasets**. Public large-scale datasets for learning-to-rank (LTR), especially in counterfactual settings, are scarce. To the best of our knowledge, we propose the first reproducible testbeds for counterfactual ranking evaluation. We utilize two existing large-scale LTR datasets: Baidu-ULTR (Hager et al., 2024; Zou et al., 2022) and Amazon KDD-Cup (Reddy et al., 2022), to construct these testbenches, enabling rigorous evaluation of permutation-aware ranking policies. Baidu-ULTR contains 1.8M query groups (11.7M query-document pairs) and 590K validation/test sessions. Amazon KDD-Cup comprises 130K queries and 2.6M annotated query-product pairs with rich textual metadata. We generate 400K training and 50K validation/test query groups by sampling permutations of products per query. See Appendix B.1 for further details.

**Implementation Details and Baselines.** Our reward models and rankers are based on a transformer architecture with 12 layers, 768 hidden dimensions, 12 attention heads, and roughly 110M parameters. We set $\tau = 0.5$ and $\lambda = 0.7$ for all REWARDRANK experiments, based on tuning over a held-out set. Ablations with varying values and further implementation details are provided in Appendix B.2. For comparison, we implement two utility-based counterfactual ranking methods: URCC (Xi et al., 2024), which uses a LambdaLoss-based pairwise objective, and PG-rank (Gao et al., 2023), which applies Plackett–Luce modeling with policy gradients. Our variants, URCC* and PG-rank*, replace their offline metric utility (e.g. NDCG) with our transformer-based reward model for improved counterfactual performance. Additionally, we train standard LTR baselines: ListNet (Cao et al., 2007), ListMLE (Xia et al., 2008), LambdaRank (Wang et al., 2018), and PiRank (Swezey et al., 2021), all using the same transformer architecture for fair comparison across supervision methods.

### 5.1 LARGE-SCALE REPRODUCIBLE TESTBENCHES FOR COUNTERFACTUAL LTR

To enable reproducible evaluation of ranking policies without online A/B testing, we introduce two complementary testbeds: PO-Eval, which leverages a parametric click model, and LAU-Eval, which simulates human-like shopping behavior via LLM reasoning. Together, they enable holistic, counterfactual assessment of ranking algorithms under both statistical and behavioral lenses.

**Parametric Oracle Evaluation (PO-Eval).** To simulate a click-based counterfactual recommendation setting, we build a testbed from the Baidu-ULTR dataset (Hager et al., 2024), employing a *pretrained parametric IPS model* as the oracle for supervision. This model estimates the click probability at position $\ell$ as $P(C) = P(E_\ell) \cdot P(R_{q,i})$, where $P(E_\ell)$ is the position-dependent examination probability and $P(R_{q,i})$ is the click probability given examination We use this oracle to sample binary clicks for training and later reuse it for counterfactual evaluation of new ranking policies. For each ranked query group (QG), we compute the expected utility as the probability of at least one click and the observed utility as a binary indicator of at least one sampled click. This setup provides a realistic and repeatable framework for evaluating how well learned rankers align with user behavior modeled by the IPS-oracle. See Appendix A.1 for details on the parametric model and the derivation of utility metrics.

Table 7 reports counterfactual evaluation results using PO-Eval, where we leverage a pre-trained parametric IPS-Oracle to simulate user clicks and assess ranking quality. The IPS-based utility $Pr(\#Clicks \geq 1)$ captures the expected probability of at least one click per ranked list, while $NDCG_{click}$ measures how high are the originally clicked items in the test dataset ranked. The *Upper-Bound* is computed by ranking items in descending order of $P(R)$, which maximizes utility due to the rearrangement inequality (Day, 1972) (see Appendix A). Traditional LTR baselines (ListNet, ListMLE, LambdaRank, PiRank), trained with per-item IPS–sampled clicks, achieve strong offline/surrogate metrics under Eq. 2 (e.g., $NDCG_{click}$) but fail to capture the true user utility in Eq. 1 (e.g., $Pr(\#Clicks \geq 1)$). URCC* yields the lowest performance, as it relies heavily on a strong *pretrained* ranker to initialize its search; without such initialization, its effectiveness diminishes significantly (see Appendix D). In particular, URCC* explores only the neighborhood of the current permutations via pairwise position swaps, which (i) induces quadratic complexity and (ii) leads to *pessimistic* exploration that can miss superior rankings outside this local region. In contrast, REWARDRANK does not require any pretrained ranker and performs counterfactual optimization directly, enabling broader exploration beyond the data rankings from logged data. For PG-Rank*, we observe that increasing the number of Monte Carlo samples (MC = 1, 5, 10) reduces variance in its estimates, which improves performance, albeit at the cost of longer training time (see Appendix

Table 1: **Counterfactual and surrogate evaluation across two settings.** The table compares ranking methods under (i) **PO-Eval** and (ii) **LAU-Eval**. For each setting we report a *counterfactual* metric: $\Pr(\#\text{Clicks} \geq 1)$ for PO-Eval and $\Pr(\#\text{Purchases} \geq 1)$ for LAU-Eval, reflecting the probability of at least one positive user action; and an *offline/surrogate* metric: $\text{NDCG}_{\text{click}}$ and $\text{NDCG}_{\text{purchase}}$, respectively, computed from logged labels (Eq. 2). While most baselines achieve high surrogate scores, these gains do not consistently translate into higher counterfactual utility (Eq. 1). Notably, $\text{URCC}^*$ and $\text{PG-rank}^*$ attain competitive NDCG yet underperform on the counterfactual metric, whereas REWARDRANK delivers the highest purchase rate in LAU-Eval while remaining competitive on surrogate metrics. The `policy_in_data` row reflects the original logged ordering.

| | **PO-Eval** | | **LAU-Eval** | |
| | *Counterfactual (✓)* | *Offline (✗)* | *Counterfactual (✓)* | *Offline (✗)* |
| **Method** | $\Pr(\#\text{Clicks} \geq 1)$ | $\text{NDCG}_{\text{click}}$ | $\Pr(\#\text{Purchases} \geq 1)$ | $\text{NDCG}_{\text{purchase}}$ |
|---|---|---|---|---|
| Upper-Bound | $0.553 \pm 0.0007$ | – | – | – |
| Policy in data | $0.475 \pm 0.0004$ | $0.211 \pm 0.0003$ | $0.497 \pm 0.009$ | $0.496 \pm 0.009$ |
| ListNet (Cao et al., 2007) | $0.523 \pm 0.0007$ | $0.376 \pm 0.0002$ | $0.521 \pm 0.009$ | $0.405 \pm 0.009$ |
| ListMLE (Xia et al., 2008) | $0.522 \pm 0.0007$ | $0.377 \pm 0.0002$ | $0.522 \pm 0.008$ | $0.402 \pm 0.008$ |
| LambdaRank (Wang et al., 2018) | $0.524 \pm 0.0007$ | $0.378 \pm 0.0002$ | $0.523 \pm 0.009$ | $0.406 \pm 0.009$ |
| PiRank (Swezey et al., 2021) | $0.525 \pm 0.0007$ | $0.378 \pm 0.0002$ | $0.528 \pm 0.007$ | $0.408 \pm 0.009$ |
| $\text{URCC}^*$ (Xi et al., 2024) | $0.462 \pm 0.0005$ | $0.315 \pm 0.0004$ | $0.471 \pm 0.008$ | $0.401 \pm 0.007$ |
| $\text{PG-rank}^*$ (Gao et al., 2023) | $0.501 \pm 0.0005$ | $0.327 \pm 0.0002$ | $0.489 \pm 0.007$ | $0.402 \pm 0.008$ |
| REWARDRANK | $\mathbf{0.536 \pm 0.0007}$ | $0.370 \pm 0.0002$ | $\mathbf{0.561 \pm 0.008}$ | $0.401 \pm 0.007$ |

Section A.3 for details). In contrast REWARDRANK attains the highest utility under IPS-Oracle, despite slightly lower $\text{NDCG}_{\text{click}}$ than some baselines. This reflects a key distinction: proxy metrics, such as NDCG (Eqn. 2), may not fully align with the true user utility (Eqn. 1). By directly optimizing counterfactual reward, REWARDRANK better aligns with behavioral objectives beyond conventional ranking accuracy.

**LLM-based User Simulation (LAU-Eval).** While PO-Eval captures position bias via IPS-Oracle supervision, it does not account for broader behavioral patterns such as brand bias, similarity aversion, or irrelevance bias. To complement PO-Eval and more fully assess human-centered ranking behavior, we introduce the LAU-Eval framework. In this setup, a large language model (LLM) is prompted to simulate user shopping behavior given a query and its associated product list from the Amazon KDD-Cup dataset. The prompt incorporates behavioral factors such as position bias, brand bias, irrelevance bias, and color bias (full details are provided in Appendix C.2). The LLM generates a binary purchase decision $D(\text{purchase}) \in \{0, 1\}$, which serves as the reward signal for training a reward model and optimizing rankers. For evaluation, the same prompt is used: each ranker's ranked item list is assesed by the LLM, and performance is reported as the average purchase decision rate on a held-out test set. For LTR methods that do not rely on reward modeling, we instead use the per-item binary LLM-purchase decision as the training signal. Higher values indicate stronger alignment with human-centered behavioral criteria. Refer to the Appendix Section C.2 for implementation details.

Under *LAU-Eval*, which measures binary purchase decisions made by the LLM, we observe clear differences across methods. The `policy_in_data` baseline (original item order) attains an average purchase rate of $0.497$. Classical listwise approaches—ListNet, ListMLE, LambdaRank, and PiRank—yield only modest gains on the true utility $\Pr(\#\text{Purchase} \geq 1)$, reaching $0.500$–$0.513$, while achieving very high scores on the offline/surrogate utility ($\text{NDCG}_{\text{purchase}}$). These LTR methods largely succeed by moving the purchased item to the top, which inflates surrogate metrics but does not faithfully capture true preferences under the LLM-Oracle, such as brand or color bias among the items, and therefore does not consistently increase purchases. This underscores the need to optimize *counterfactual utility* as the primary metric for modeling human ranking behavior. We also observe a clear mismatch between surrogate and counterfactual objectives for counterfactual baselines: both $\text{PG-rank}^*$ and $\text{URCC}^*$ attain a strong $\text{NDCG}_{\text{purchase}}$ (formulated by Eqn 2), yet *both* methods yield lower values on the counterfactual metric (purchase rate as formulated by Eqn 1). This indicates that optimizing the ranking-aware surrogate alone can overfit to list reshuffling (e.g., moving a known purchased item to the top) without improving the actual decision outcome measured by $\Pr(\#\text{Purchase} \geq 1)$.

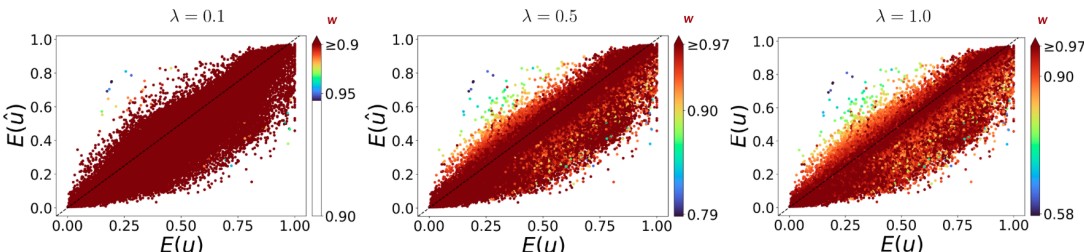

Figure 3: **Reward misspecification correction on PO-Eval.** Each point represents a ranked list with true utility (u: estimated by IPS-Oracle) and predicted utility ($\hat{u}$: estimated by utility model) from the ranker. Colors indicate ($w = 1 - \lambda|u_{\text{logged}} - \hat{u}_{\text{logged}}|$), showing how increasing ($\lambda$) down-weights overconfident or misaligned samples to emphasize well-calibrated predictions.

**Ablations.** We ablate the per-item regularizer and the two parameters of REWARDRANK: the SoftSort temperature $\tau$, which controls the sharpness of the permutation approximation (8), and the misspecification correction strength $\lambda$, which down-weights rewards on QGs with high prediction error (13). Removing the auxiliary item-level reward loss (Eqn 6) decreased the final expected counterfactual utility of the learned ranker. This indicates that learning to predict the per-item feedback enhances the reward model's generalization and hence improves downstream ranking performance. As shown in Figure 3, increasing $\lambda$ progressively reduces the influence of unreliable reward estimates by lowering their instance weights, leading to more stable learning. As $\lambda$ increases, the influence of low-confidence predictions (lower $w$) diminishes, effectively down-weighting misspecified instances. This correction improves stability by emphasizing samples with well-aligned predicted rewards. For illustration, we display soft utility scores from Eqn 17; however, all experiments use binary utility signals as defined in Eqn 16. We find that $\tau = 0.5$ and $\lambda = 0.7$ achieve the best trade-off between stability and performance. Full details of these ablations are reported in Appendix D.

### 5.2 BAIDU-ULTR DATASET WITH REAL USER CLICKS

While we previously used the Baidu-ULTR dataset within the PO-Eval framework under IPS-Oracle supervision, here we instead rely directly on the real click signals provided in the data. Following the protocol in (Hager et al., 2024), models are trained with a binary click labels for each query groups ($u = 1$ if any item is clicked, otherwise 0). Since counterfactual evaluation is not feasible here, we follow (Hager et al., 2024) and report *Relevance DCG* at 5 and 10, computed on the human-assigned relevance labels provided with the test set[1]. As shown in Table 2, our method achieves a new state of the art $DCG@5$ and $DCG@10$ across all baselines. Importantly, these improvements are observed on human-assigned relevance labels that were never used in training by any method. This is particularly noteworthy given that our method is not optimized for relevance DCG. These results highlight both the robustness of our approach and its ability to generalize to real human feedback in large-scale search settings.

Table 2: **Baidu-ULTR with real clicks.** RE-WARDRANK achieves SOTA performance. [†] metrics are taken from Hager et al. (2024)

| Method | $DCG_{\text{rel}}@5$ | $DCG_{\text{rel}}@10$ |
| --- | --- | --- |
| Point IPS[†] (Hager et al., 2024) | 4.79 | 7.43 |
| List IPS[†] (Hager et al., 2024) | 5.20 | 7.88 |
| LambdaRank[†] (Hager et al., 2024) | 5.45 | 8.23 |
| ListNet (Cao et al., 2007) | 5.05 | 7.64 |
| ListMLE (Xia et al., 2008) | 5.13 | 7.88 |
| PiRank (Swezey et al., 2021) | 5.23 | 8.01 |
| URCC[*] Xi et al. (2024) | 5.01 | 7.44 |
| PG-rank[*] Gao et al. (2023) | 5.09 | 7.62 |
| REWARDRANK | **5.83** | **8.42** |

## 6 CONCLUSION

We present REWARDRANK, a counterfactual ranking framework that directly optimizes a behaviorally grounded utility instead of relying on proxy click-based surrogates. Notably, our approach accom-

---

[1]We report DCG rather than NDCG for consistency with (Hager et al., 2024)

plishes this without imposing any explicit modeling assumptions. Architecturally, REWARDRANK uses *SoftSort* to produce a differentiable soft permutation matrix, enabling end-to-end learning with *soft item embeddings* (convex combinations over items) that feed a utility model. To guard against reward model misspecification, we include a *misspecification regularization* term which is an explicit $\lambda$-weighted correction that penalizes over-reliance on noisy preference signals and stabilizes updates against spurious gains. Through the proposed *PO-Eval* and *LAU-Eval* protocols, we showed a systematic mismatch between offline/surrogate metrics (e.g., $\text{NDCG}_{\text{purchase}}$) and true decision outcomes, and demonstrated that REWARDRANK achieves the highest purchase rates while remaining competitive on surrogate metrics. Unlike URCC*, REWARDRANK does *not* require a pretrained ranker and can leverage sessions without purchase labels, extracting useful signal in sparse-feedback regimes. Ablations further indicate that auxiliary per-item losses (including on purchase-free QGs) provide consistent, moderate gains. Overall, aligning training and evaluation with counterfactual utility yields models that better capture decision-relevant user behavior than traditional LTR or locally exploratory counterfactual baselines.

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

# A  PROOFS AND CONCEPTUAL DETAILS

## A.1  CLICK-BASED UTILITY FOR PO-EVAL.

The IPS-Oracle simulates user clicks as a probabilistic function of both position-dependent examination and item-specific relevance. Specifically, the click probability for item $x_{\pi(\ell)}$ at position $\ell$ under ranking $\pi$ is modeled as:

$$P(C_{q,x_{\pi(\ell)},\ell}) = P(E_\ell) \cdot \sigma(R_{q,x_{\pi(\ell)}}) \tag{14}$$

where $P(E_\ell)$ denotes the examination probability at position $\ell$, and $\sigma(R_{q,x_{\pi(\ell)}})$ is the probability of a click given examination. Given a query group $(q, \{x_\ell\}_{\ell=1}^L, \pi)$, the click indicator for each item is sampled as:

$$c_{q,x_{\pi(\ell)},\ell} \sim \text{Bernoulli}\big(P(C_{q,x_{\pi(\ell)},\ell})\big) \tag{15}$$

We define the *group-level utility* under the logged policy as a binary signal indicating whether at least one item in the list was clicked:

$$U(q, \{x_\ell\}, \pi) = \begin{cases} 1, & \text{if } \sum_{\ell=1}^L c_{q,x_{\pi(\ell)},\ell} > 0, \\ 0, & \text{otherwise.} \end{cases} \tag{16}$$

The corresponding observed utility in the dataset, $u$, is a realization of $U(q, \{x_\ell\}, \pi_{\text{log}})$ under the logged ranking $\pi_{\text{log}}$.

To obtain a differentiable approximation, we define the expected probability of at least one click as:

$$U_{\text{IPS}}(q, \{x_\ell\}, \pi) = 1 - \prod_{\ell=1}^L \big(1 - P(E_\ell) \cdot \sigma(R_{q,x_{\pi(\ell)}})\big) \tag{17}$$

This smoothed utility represents the expected engagement for ranking $\pi$ and serves as a continuous training signal. The reward model is trained to predict the binary group-level utility $u \in \{0, 1\}$ from the logged policy, while the ranker maximizes the expected soft utility $U_{\text{IPS}}$ under its own predicted rankings. This formulation bridges synthetic click modeling with realistic counterfactual feedback, enabling effective utility-based optimization even without direct supervision on full permutations.

## A.2  IDEAL IPS-ORACLE: REARRANGEMENT INEQUALITY

**Theorem 1** (Ideal Ranking Maximizes Utility via Rearrangement Inequality). *Let $\mathbf{r} = (r_1, \ldots, r_n) \in \mathbb{R}_{\geq 0}^n$ be a vector of predicted relevance scores, and let $\mathbf{e} = (e_1, \ldots, e_n) \in \mathbb{R}_{\geq 0}^n$ be a non-increasing sequence of examination probabilities: $e_1 \geq e_2 \geq \ldots \geq e_n$. Let $\pi^*$ be the permutation that sorts $\mathbf{r}$ in descending order: $r_{\pi^*(1)} \geq r_{\pi^*(2)} \geq \ldots \geq r_{\pi^*(n)}$. Then, for any permutation $\pi \in \mathcal{S}_n$, we have:*

$$\sum_{i=1}^n e_i \cdot r_{\pi^*(i)} \geq \sum_{i=1}^n e_i \cdot r_{\pi(i)}$$

*Proof.* This is a direct consequence of the classical rearrangement inequality (Day, 1972). Among all permutations $\pi$ of the relevance scores, the weighted sum $\sum_i e_i \cdot r_{\pi(i)}$ is maximized when the $r_{\pi(i)}$ are ordered in the same way as the $e_i$, i.e., both decreasing. Hence, sorting $\mathbf{r}$ in descending order and aligning it with the already sorted $\mathbf{e}$ gives the maximal utility. □

Above analysis shows that ideal ranking order under the IPS Oracle is ordering the items such the sorting of item relevance scores and examination probabilities result in the same permutation.

## A.3  DETAILS OF BASELINES

### A.3.1  PG-RANK$^*$ : PG-RANK WITH LEARNED REWARD MODEL.

We extend the PG-Rank framework (Gao et al., 2023) by replacing the handcrafted reward (e.g., NDCG) with a learned reward model $g(q, \{i\}_L, \pi)$ that scores entire permutations based on user

utility. The goal is to maximize the expected reward under the Plackett–Luce distribution induced by the ranker's scores:

$$\mathcal{L}_{\text{PG-reward}}(\theta) = \mathbb{E}_{\pi \sim \mathbb{P}_\theta} \left[ g(q, \{i\}_L, \pi) \right] \tag{18}$$

where $\mathbb{P}_\theta(\pi)$ is the Plackett–Luce distribution over permutations, parameterized by model scores $s_1, \ldots, s_L$ for each item in the query group. To enable backpropagation through the sampled permutations, we adopt the Gumbel-Softmax trick as in the original PG-Rank implementation, which provides a continuous relaxation of the discrete sampling process.

The gradient of this objective is estimated using the REINFORCE trick with a baseline $b$ for variance reduction (adopted from PG-rank (Gao et al., 2023; Kool et al., 2019)):

$$\nabla_\theta \mathcal{L}_{\text{PG-reward}} \approx \frac{1}{K} \sum_{k=1}^{K} \left[ \left( g(\pi^{(k)}) - b \right) \cdot \nabla_\theta \log \mathbb{P}_\theta(\pi^{(k)}) \right] \tag{19}$$

where $\pi^{(k)} \sim \mathbb{P}_\theta$ are $K$ Monte Carlo samples drawn from the Plackett–Luce model.

The log-probability of a sampled permutation $\pi$ under this model is given by:

$$\log \mathbb{P}_\theta(\pi) = \sum_{k=1}^{L} \left[ s_{\pi(k)} - \log \sum_{j=k}^{L} \exp(s_{\pi(j)}) \right] \tag{20}$$

This formulation allows us to train the ranking model directly on learned, utility-aligned reward signals using fully differentiable, sample-based policy gradients.

### A.3.2 URCC* WITH LEARNED REWARD MODEL.

URCC (Xi et al., 2024) proposes a two-stage counterfactual reranking framework that jointly learns a set-aware utility function and a context-aware reranker. The utility model in URCC is itself learned from data and used to guide the optimization of the reranker via a pairwise ranking loss over permutations. Since the official implementation of URCC is not publicly available, we re-implemented the method using our own architecture.

In our version of URCC*, we retain the core two-stage structure but implement the utility model $g(q, \{i\}_L, \pi)$ as a Transformer-based encoder trained to predict user utility over full permutations. Given a query $q$ and a set of items $\{i\}_L$, the reward model assigns a scalar score to a permutation $\pi$:

$$S_\pi = g(q, \{i\}_L, \pi) \tag{21}$$

Following URCC, we then train the ranker $f_\theta$ to maximize this learned reward by optimizing a context-aware pairwise loss. For a pair of permutations $(\pi^+, \pi^-)$ such that $g(q, \{i\}_L, \pi^+) > g(q, \{i\}_L, \pi^-)$, we minimize the following objective:

$$\mathcal{L}_{\text{URCC-reward}}(\theta) = \mathbb{E}_{(\pi^+, \pi^-) \sim \mathcal{P}} \left[ \log \left( 1 + \exp \left( -(S_{\pi^+} - S_{\pi^-}) \right) \right) \right] \tag{22}$$

Here, $\mathcal{P}$ denotes the set of sampled permutation pairs with preference orderings induced by the reward model. Our implementation uses neighborhood-based sampling (e.g., pairwise swaps) to construct $\pi^+$ and $\pi^-$ from the base ranking.

Thus, while our training procedure is structurally consistent with the original URCC framework, we employ a more expressive Transformer-based reward model to capture user behavior better and align optimization with utility-oriented objectives.

### A.4 COMPARISON OF TIME COMPLEXITY AND COUNTERFACTUAL SPACE EXPLORATION

Table 3 compares the time complexity of three methods: URCC*, PG-rank*, and REWARDRANK. The per-iteration time complexity is analyzed based on the number of calls to the reward model.

Table 3: **Comparison of Time Complexity** for URCC* , PG-rank* , and REWARDRANK in term of number of calls to the reward model per iteration on Baidu-ULTR dataset.

| Method | Time Complexity | Wall-Clock Time | Description |
|---|---|---|---|
| PiRank | 1 | $\sim$6 hours | No call to the reward model |
| URCC* | $n^2$ | $\sim$34 hours | Neighborhood search, pessimistic |
| PG-rank* | $k$ | $\sim$16 hours ($k = 10$) | Needs large $k$ for convergence |
| RewardRank | 1 | $\sim$7 hours | Full counterfactual space exploration |

- **URCC* :** $n^2$, where $n$ is the number of items in the list. URCC* explores the neighborhood of factual permutations, leading to quadratic complexity due to pairwise comparisons. URCC* only explores the neighborhood of factual permutations, meaning it performs limited counterfactual exploration. This approach is considered pessimistic because it does not explore the entire space of possible rankings, which could miss potentially better arrangements.

- **PG-rank* :** $k$, where $k$ is the number of Monte Carlo (MC) samples. While $k$ is typically smaller than $n$, PG-rank* requires large $k$ values and variance reduction baselines to converge. PG-Rank uses Monte Carlo (MC) sampling to explore a broader counterfactual space, but this approach requires large MC samples to converge effectively. To ensure stable and accurate exploration, PG-Rank relies on variance-reduction baselines. However, it still faces challenges in accurately capturing all potential counterfactual configurations without a very large number of samples.

- **REWARDRANK:** 1, as it performs a single call to the reward model. RewardRank explores the entire counterfactual space efficiently and can focus on more certain regions with reward misspecification mitigation.

In Table 3, we also provide the overall wall-clock time to train the model under the above method for the Baidu-ULTR dataset. Each model is trained for 21 epochs.

### A.5 EXTENDED RELATED WORK

**Unbiased Learning to Rank (ULTR).** Unbiased Learning to Rank aims to correct biases in user interaction data such as position bias and examination bias. Classical methods rely on inverse propensity weighting or dual learning schemes to debias logged clicks Joachims et al. (2017); Ai et al. (2018). Model-based ULTR introduces explicit click models to recover unbiased relevance estimates Luo et al. (2020). A recent survey provides a comprehensive comparison of online and offline ULTR frameworks Ai et al. (2021), and more recent work applies distributionally robust optimization to improve stability under click-model misspecification Niu et al. (2025a). A reproducibility study further examines the robustness of counterfactual LTR methods under various click models Niu et al. (2025b). These ULTR methods focus primarily on correcting biased *relevance* signals under specific click models, whereas our work aims to learn and optimize a *utility* function that captures list-level behavior beyond pointwise relevance.

**Reinforcement Learning To Rank (RLTR).** RL-based ranking methods optimize listwise rewards using stochastic policies and importance weighting. Coarse-grained RLTR methods learn from session-level reward signals Tu et al. (2022), and other approaches apply policy gradients to optimize top-$k$ or click-based objectives Chen et al. (2019). However, RLTR methods require behavior-policy estimation, suffer from high variance, and cannot evaluate permutations outside the support of the logged policy. In contrast, our framework evaluates arbitrary permutations deterministically through SoftSort, without requiring a stochastic policy or behavior-policy estimation.

**Utility-Based and List-Level Modeling.** Utility-based or list-level modeling has a long history in learning-to-rank, especially in RL-based frameworks that optimize listwise reward signals directly Tu et al. (2022). Beyond classical RLTR, several recent approaches focus explicitly on modeling list-level interactions, such as multi-generator reranking systems that capture global list structure Yang et al. (2025) and utility-oriented reranking models that aim to directly optimize user utility rather

than relevance Xi et al. (2024). Policy-gradient–based ranking approaches such as PG-Rank Gao et al. (2023) and PiRank Swezey et al. (2021) similarly optimize listwise objectives using differentiable sorting or sampled permutations. Other differentiable listwise frameworks—such as NeuralNDCG Pobrotyn & Białobrzeski (2021), stochastic relaxations of sorting networks Grover et al. (2019), and permutation-level re-ranking systems Shi et al. (2023); Feng et al. (2021); Ren et al. (2024)—also emphasize modeling dependencies across the entire slate.

In counterfactual settings, recent work has explored utility-aware optimization under safety constraints Gupta et al. (2024b;a) and introduced affine or doubly-robust corrections to improve stability of listwise estimators Vardasbi et al. (2020); Saito & Joachims (2022). However, these methods typically rely on stochastic exploration, importance weighting, or sampling-based reranking to traverse the space of permutations. In contrast, our approach learns a parametric utility model that can score arbitrary counterfactual permutations and couples this with deterministic SoftSort-based optimization. This enables full-permutation exploration without the variance, behavior-policy dependence, or sampling overhead characteristic of RL-based and counterfactual reranking approaches.

**Transformer-Based Ranking Models.** Transformer architectures have become standard in large-scale retrieval, re-ranking, and recommendation systems. Modern neural rankers such as Non-Autoregressive Re-ranking models Ren et al. (2024), UniRank Pei et al. (2021), DASALC Qin et al. (2021), DeepRank Pang et al. (2017), and MatRank Luo et al. (2022) leverage self-attention or attention-inspired mechanisms to capture rich query–item and item–item interactions. These models typically optimize relevance-oriented objectives or matching scores at the item level. Our work differs in focus: instead of modeling pointwise or pairwise relevance, we use a transformer to parameterize a list-level *utility* function and combine it with a differentiable ranking operator, allowing direct optimization toward counterfactual utility rather than traditional relevance-based objectives.

**Counterfactual Evaluation Protocols.** ULTR research often evaluates on BaiduULTR using simulated click models such as PBM, DCM, and cascade models Ai et al. (2021); Niu et al. (2025b). These simulations test bias correction under predefined user-behavior assumptions. In contrast, PO-Eval provides a parametric oracle for evaluating counterfactual utility, and LAU-Eval uses LLM-based list-level assessments to capture behavioral preferences (e.g., redundancy aversion, brand consistency) that cannot be expressed with pointwise human labels. These evaluation methods complement, rather than replace, traditional ULTR simulations by focusing on list-level *utility* rather than click-model fidelity.

While prior model-based ULTR and coarse-grained RL methods also learn reward estimates, they differ from our approach in key ways. Model-based ULTR focuses on bias-correcting click signals and then optimizes standard relevance-based objectives, whereas RewardRank learns a permutation-aware utility function that captures list-level behavioral effects beyond relevance and directly optimizes this utility through a differentiable soft-permutation operator. Coarse-grained RL approaches still rely on stochastic policies and importance weighting, limiting them to permutations explored by the behavior policy; RewardRank removes this dependence entirely by using a deterministic SoftSort-based optimization that can evaluate and optimize any permutation. Finally, prior ULTR methods optimize traditional metrics on BaiduLTR, while RewardRank introduces a misspecification-robust objective and two counterfactual evaluation suites (PO-Eval and LAU-Eval). These components provide capabilities not present in existing work and lead to consistent improvements in our experiments.

# B EXPERIMENTATION DETAILS

## B.1 DATASETS

**Baidu-ULTR Reranking Dataset.** The Baidu-ULTR dataset (Hager et al., 2024), a large-scale subset of the Baidu-ULTR corpus (Zou et al., 2022), contains user click interactions over web search queries. It includes 1.8M query groups (11.7M query-document pairs) and 590K validation/test sessions (4.8M pairs).The authors of (Hager et al., 2024) provide BERT-based CLS embeddings for each query-document pair.

We use the large-scale reranking dataset introduced by (Hager et al., 2024): publicly available at: `https://huggingface.co/datasets/philipphager/baidu-ultr_`

`uva-mlm-ctr`, derived from the original Baidu-ULTR corpus (Zou et al., 2022). This dataset is constructed from real-world user interactions on Baidu's production search engine and is designed to support robust evaluation of learning-to-rank models in counterfactual settings.

Each session consists of a user query, a candidate list of documents retrieved by an upstream ranker, the original presented ranking, and user interaction logs (e.g., clicks and dwell time). For each query-document pair, the dataset provides both sparse lexical features (e.g., BM25, TF-IDF, query likelihood) and dense semantic representations.

To generate the dense features, the authors pretrain a BERT-style model, referred to as MonoBERT, from scratch using masked language modeling (MLM) on the full Baidu corpus. This model is trained in a mono-encoder configuration and outputs a [CLS] token embedding for each query-document pair. These CLS embeddings are included in the dataset and serve as fixed, high-quality dense features for downstream reranking. The pretrained MonoBERT model and inference code are publicly available at: `https://github.com/philipphager/baidu-bert-model`.

**Amazon KDD-cup.** The KDD-Cup dataset (Reddy et al., 2022) contains 130K queries and 2.6M annotated query-product pairs in English, Japanese, and Spanish. Each query is linked to up to 40 products with rich textual metadata (titles, descriptions, bullet points), making it well-suited for LLM-based evaluation, unlike Baidu-ULTR. Although the presentation order is not recorded, the dataset primarily consists of relevant query-product pairs that were shown to users. For training, validation, and testing, we sample five random permutations of length 8 per query, resulting in 400,000 training and 50,000 validation/test groups. We use the English subset of the product search dataset released as part of the KDD Cup 2022 challenge (Reddy et al., 2022), which contains real-world queries and associated candidate products from Amazon. Each query-product pair is annotated using the ESCI labeling scheme: **E**xact match, **S**ubstitute, **C**omplement, or **I**rrelevant.

Each query group is identified by a unique `query_id` and paired with 10–40 product candidates. For each product, the dataset provides structured metadata including:

- product_title,
- product_brand,
- product_color
- product_description,
- product_bullet_point (optional fields)
- product_id,
- product_locale, and
- ESCI relevance label

To construct our training and evaluation sets, we sample 5 random permutations of length 8 from each query group. Note that we do not use the human-annotated ESCI labels provided in the dataset. Instead, we leverage the LLM's capability for contextual understanding to generate relevance labels automatically. Ideally, the relevance judgments produced by the LLM should align closely with those of human annotators. This yields approximately 392K query groups for training and 20K for validation, and 20K for testing. For a given query group, we encode each query-item pair into sentence embeddings using the `all-MiniLM-L6-v2`: `https://huggingface.co/sentence-transformers/all-MiniLM-L6-v2` model from Sentence Transformers. The input format for the sentence transformer is constructed as:

Table 4: **Statistics of the Baidu-ULTR reranking dataset** (Hager et al., 2024).

| Split | #Query Groups | #Query-Document Pairs |
|---|---|---|
| Training | 1,857,248 | 11,738,489 |
| Validation/Test | 590,612 | 4,797,378 |
| **Total** | 2,447,860 | 16,535,867 |

```
{query} [SEP] {product_title} Brand:{brand}
Color:{color}
```

Table 5: **Statistics of the Amazon KDD Cup (ESCI) dataset (English subset).**

| Split | #Query groups | #Query-Product Pairs |
|---|---|---|
| Training | 78,447 | 627,576 |
| Validation | 4,000 | 32,000 |
| Test | 4,000 | 32,000 |
| **Total** | 86,447 | 691,576 |
| **Total (including 5 random permutations)** | 392,235 | 3,137,880 |

### B.2 IMPLEMENTATION DETAILS

We use a transformer architecture for both the reward model and the ranker across all methods to ensure a consistent architectural backbone. The model contains 12 transformer layers, 768 hidden dimensions, 12 attention heads, and approximately 110M parameters. All models are trained with a learning rate of $2 \times 10^{-5}$ using the AdamW optimizer (Loshchilov & Hutter, 2017) with a weight decay of $10^{-2}$. We use a batch size of 512 and train for 21 epochs, applying a learning rate decay at epoch 12 via a step-based learning rate scheduler. All experiments are conducted using 2 NVIDIA A100 GPUs (40GB each).

For our method, REWARDRANK, we use a soft permutation temperature $\tau = 0.5$ and reward correction term $\lambda = 0.7$. In the PG-rank* baseline, which replaces the handcrafted NDCG utility with our learned reward model, we apply Gumbel-Softmax sampling with temperature 0.1 to approximate permutation sampling from the Plackett–Luce distribution. We report PG-rank* results for different Monte Carlo samples (MC = 1, 5, 10) to evaluate variance in reward estimation.

In our URCC* implementation, we follow the original two-stage design: a set-aware utility model and a pairwise ranker. The utility model is trained with a binary cross-entropy loss computed over per-item logits derived from the transformer encoder outputs. Specifically, for each item in the permutation, we pool its embedding from the encoder, apply dropout, and project it through a shared per-item classifier. The per-item predictions are matched to click labels, and their aggregated loss forms the utility supervision.

As an additional baseline, we include a Naive-ranker trained with a relaxed NDCG objective following the PiRank formulation (Swezey et al., 2021), allowing listwise supervision using soft permutation matrices. All baselines are trained using the same reward data and input embeddings to isolate the impact of the learning objective.

Representative code for our implementations of REWARDRANK, PG-rank* , URCC* baselines, and evaluation procedures is included in the supplementary material.

## C COUNTERFACTUAL EVALUATION PROTOCOLS

### C.1 PO-EVAL DETAILS

PO-Eval provides a click-based framework for counterfactual evaluation of ranking models. Using the pre-trained Inverse Propensity Scoring model (IPS-Oracle) (Hager et al., 2024)[2] on the Baidu-ULTR dataset, it generates soft click probabilities for items in a ranked query group. These probabilities serve as counterfactual labels, enabling the evaluation of how effectively a ranker can model user engagement patterns reflected in clicks.

As the Baidu-ULTR dataset is derived from user interaction logs, click activity is heavily concentrated in the top-ranked positions, reflecting strong position bias (see Figure 4b). In contrast, the distilled soft utility ($\hat{y}$) generated by the IPS-Oracle exhibits a more uniform distribution across positions

---

[2] https://github.com/philipphager/baidu-bert-model

(Figure 4a), indicating that the oracle has successfully learned to correct for position bias. Under the PO-Eval protocol, ranking methods aim to implicitly learn position debiasing from the IPS-Oracle's soft utility, as indicated by high $U_{\text{IPS-O}}(q, \{i\}_L, \hat{\pi})$.

**Training and evaluating ranking schemes.** Using the learned reward model, any ranker $f$ can be optimized via the reward maximization objective defined in Eqn 12. To evaluate its performance under the IPS-Oracle, we define the following metric: Given a query group $(q, \{i\}_L)$ and predicted relevance scores $s = [s_1, \ldots, s_L]$, the induced permutation is $\hat{\pi} = \text{argsort}(s)$. For each position $\hat{\pi}_\ell$, the examination probability is $P(E_{\hat{\pi}_\ell})$, and the associated relevance score $R_{q,i_\ell}$ is provided by the IPS-Oracle. The overall utility is computed as the probability of at least one click: $U_{\text{IPS}}(q, \{i\}_L, \hat{\pi})$, which serves as the primary evaluation metric. It reflects how well $f$ aligns with the user behavior modeled by the IPS-Oracle; higher values indicating better alignment. Additionally, we report $\text{NDCG}_{\text{rel}}@10$, which measures how much the predicted ranking respects the relevance scores $R_{q,i}$.

We incorporate the examination probabilities from (Hager et al., 2024), which are defined as:

$$P(E) = \{1 : 1.0000, 2 : 0.6738, 3 : 0.4145, 4 : 0.2932, 5 : 0.2079, 6 : 0.1714, 7 : 0.1363, 8 : 0.1166\}$$

## C.2 LAU-Eval DETAILS

We use Claude 3.5 Sonnet v2 with a temperature of 0.5 and a context window of 5,000 tokens. The LLM is prompted using a consistent instruction template, as illustrated in Figure 6. To evaluate a ranker with LAU-Eval, its predicted scores are converted into item positions, which are then used to reorder the input list. This reordered list is passed to the LLM alongside the original query, and the LLM outputs a binary decision regarding purchase. We include representative query groups and the corresponding LLM responses to demonstrate this pipeline.

To assess the robustness of LAU-Eval under different sampling conditions, we examine how varying the temperature of the LLM decoding process affects its outputs. Figure 5 shows the distributions of LLM-simulated purchase decisions and selected item positions at temperatures 0.1, 0.5, and 0.75. While purchase rates exhibit slight variation, the LLM consistently favors top-ranked items—reflecting realistic user behavior in shopping scenarios.

**Instruction prompt for LLM.** We design the LLM-Eval instruction to incorporate behavioral biases such as position bias, brand preference, irrelevance filtering, similarity aversion, and color bias, guiding the LLM to consider both relevance and context-dependent preferences. Given a query and an ordered product list, the LLM estimates (i) the probability of purchasing at least one item and (ii) the selected item, without explicit relevance constraints. We illustrate the instruction prompt using an example from the Amazon KDD-Cup dataset (Reddy et al., 2022), as shown in Figure 6.

**Ranking Evaluations.** We present the LLM's response to the initial list in Figure 7, including the full reasoning behind the response. It is noteworthy how the LLM is able to reason about the biases

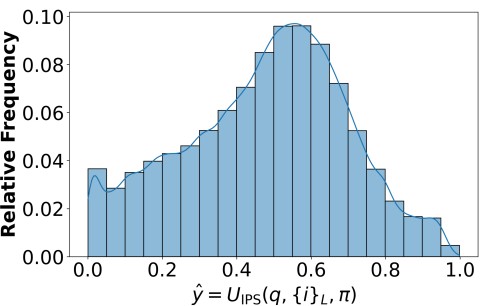

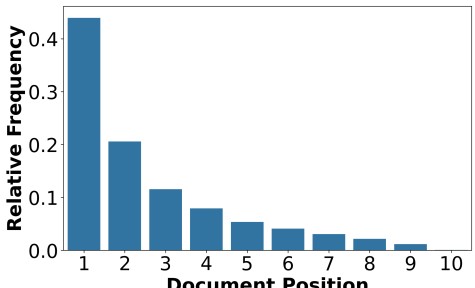

(a) Soft-utility distribution on Baidu-ULTR generated by IPS-Oracle computed using Eqn 17.

(b) Position distribution of clicks in Baidu-ULTR.

Figure 4: Distributions extracted from IPS-Oracle analysis on Baidu-ULTR.

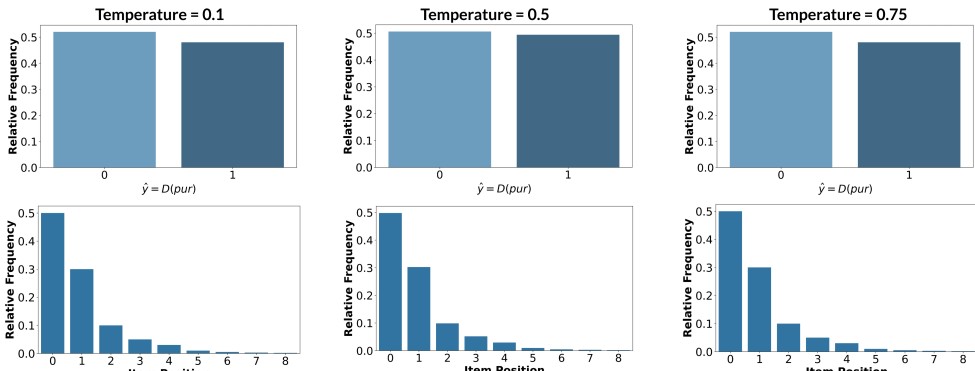

Figure 5: **Effect of Sampling Temperature on LLM-Simulated Behavior in LAU-Eval.** We visualize the distribution of binary purchase decisions (top) and item positions (bottom) generated by Claude Sonnet 3.5 v2 under three sampling temperatures: 0.1, 0.5, and 0.75. Each sample corresponds to a ranked list generated during LAU-Eval. As temperature increases, the purchase signal slightly diversifies, while positional biases remain consistent across settings. These results suggest that LAU-Eval is robust to moderate sampling variability, with LLMs producing stable user-like behavior under soft prompting.

Table 6: **Counterfactual vs. surrogate evaluation of ranking methods (LAU-Eval only).** We report performance on the true counterfactual utility (Eq. 1) and offline/surrogate metrics (Eq. 2). While most methods score highly on surrogate metrics, these gains often fail to align with true user utility.

| | **LAU-Eval** | | |
| | *Counterfactual (✓)* | *Offline (✗)* | |
| **Method** | $\Pr(\#\text{Purchases} \geq 1)$ | $\text{NDCG}_{\text{purchase}}$ | $\text{NDCG}_{\text{ESCI}}$ |
| Policy in data | $0.497 \pm 0.009$ | $0.496 \pm 0.009$ | $0.995 \pm 0.009$ |
| ListNet (Cao et al., 2007) | $0.521 \pm 0.009$ | $0.405 \pm 0.009$ | $0.8611 \pm 0.009$ |
| ListMLE (Xia et al., 2008) | $0.522 \pm 0.008$ | $0.402 \pm 0.008$ | $0.8610 \pm 0.003$ |
| LambdaRank (Wang et al., 2018) | $0.523 \pm 0.009$ | $0.406 \pm 0.009$ | $0.8610 \pm 0.009$ |
| PiRank (Swezey et al., 2021) | $0.528 \pm 0.007$ | $0.408 \pm 0.009$ | $0.8623 \pm 0.005$ |
| URCC* (Xi et al., 2024) | $0.471 \pm 0.008$ | $0.401 \pm 0.007$ | $0.8621 \pm 0.009$ |
| PG-rank* (Gao et al., 2023) | $0.489 \pm 0.007$ | $0.402 \pm 0.008$ | $0.8630 \pm 0.009$ |
| REWARDRANK | $0.561 \pm 0.008$ | $0.401 \pm 0.007$ | $0.8628 \pm 0.009$ |

present in the query groups effectively. For each initial list, we also show the LLM's response to the rearranged list generated by Claude, depicted in Figure 8. As seen, the initial arrangements in Figure 7 lead to a no purchase decision, whereas REWARDRANK generates arrangements that increase the likelihood of purchase according to the LLM. Furthermore, the LLM's response enhances the interpretability of LLM-Eval, demonstrating how REWARDRANK's ranking capabilities align with the LLM's reasoning process.

Initially, we experimented with smaller language models such as Llama-3.1-8B: `meta-llama/Llama-3.1-8B-Instruct` and DeepSeek-R1-Distill: `deepseek-ai/DeepSeek-R1-Distill-Llama-8B`. However, these models were unable to generate appropriate responses to the instructions. Our experiments revealed that larger models were better at understanding the context.

It is important to note that LAU-Eval is used to simulate user behavior dynamics that may influence user decisions. Our selection of biases and instruction prompt serves as a proof-of-concept demonstrating that an LLM can be used as a proxy user to study counterfactual ranking strategies. We

acknowledge that there are likely many variants of instruction prompts that could be designed to simulate user behavior. This area of exploration could be a direction for future work.

**LLM Prompt (Probability Estimation Task)**

You are shopping for a given query. Your task is to estimate the likelihood of purchasing any item in a provided list. Please answer yes or no, indicating whether you wish to purchase any item from the given list. Consider the relative relevance of items in the list when making your decisions. Be frugal, as a typical human user would be—most users buy when the list is highly relevant, and often make no purchase when following behavioral criteria are not met. You enter a 'query' into the shopping system, and it returns some items mentioned in the 'products'. The items are presented in the given order, with 1st item shown at the top of the list and the last item shown at the bottom.

Your query-products shopping list:

**Query:** `11 iphone pro screen protector`

**Products:**

```
{"B09CGJ8RW1":"title":"JETech Screen Protector and Camera Lens",
            "brand":"JETech","color": "Transparent",
"B07515P7PT":"title":"JETech Screen Protector for iPhone 11 Pro",
            "brand":"JETech","color": "Clear",
"B075S8V728":"title":"Ailun for Apple iPhone 11 Pro/iPhone",
            "brand":"Ailun","color":NA,
"B07STC633H":"title":"UNBREAKcable Screen Protector for iPhone 11",
            "brand":"UNBREAKcable","color":NA,
"B07D6XR7FM":"title":"TETHYS Glass Screen Protector for iPhone 11",
            "brand":"TETHYS","color":"Transparent",
"B073DLZWX7":"title":"Maxboost Screen Protector for Apple iPhone",
            "brand":"Maxboost","color":"Clear",
"B07FP41MC5":"title":"Trianium (3 Packs) Screen Protector",
            brand:"Trianium",color:"Clear",
"B09BQRWG15":"title":"YRMJK Screen Protector Compatible iPhone",
            brand:"YRMJK",color: NA}
```

**Relevance Score**: The relevance score shows how relevant the item is given the query. For every query-item pair, it is a numerical value between 0 and 1. You should consider the following criteria:

1. **Position bias**: where the items appearing near the top are more likely to be clicked. The position score decreases based on the following examination probabilities: position_scores = { 1: 1.0000, 2: 0.6738, 3: 0.4145, 4: 0.2932, 5: 0.2079, 6: 0.1714, 7: 0.1363, 8: 0.1166 } If the relevant item is not near the top, it will reduce the probability of purchase irrespective of its relevance.

2. **Brand bias**: If items from the same brand are placed adjacent to each other, the user is less likely to make a purchase. High brand bias means adjacent items are from the same brand.

3. **Irrelevance bias**: Multiple irrelevant items near the top reduce the chance of purchasing any item. This measures contextual dissimilarity at top positions.

4. **Color bias**: A lack of visual diversity (e.g., similar-colored items grouped together) discourages purchases.

Note that high brand-bias, irrelevance-bias, or color-bias harms the user experience and should lower the chance of purchasing. Act like a frugal user: only purchase if all criteria are satisfied; otherwise, do not purchase.

**Task**: Given a query and its candidate product list, decide whether any item should be purchased and report the decision (no exact totals needed). Use a strict policy: purchase only if all criteria are satisfied; otherwise, output no purchase. **Decision rule**: Determine the Final decision (yes/no) by jointly considering relevance score, position bias, brand bias, irrelevance bias, and color bias. Proceed to purchase only if all criteria are satisfactorily met. If not, do not purchase. Be especially cautious about brand and color biases—purchase only when the biases are very low. If the decision is yes, select exactly one item to purchase.

**Output**: D(purchase) = <answer>

If decision = yes, then Item to be purchased = <answer>

Figure 6: **LLM instructions for a sample taken from the Amazon KDD-Cup dataset (Reddy et al., 2022)**. A total of 8 products are displayed for the query "*11 iphone pro screen protector*". For visual clarity, we have shortened the titles and product descriptions.

---

**LLM's response to the initial item list**

Let me analyze this shopping scenario step by step:
1. **Relevance Analysis**:- Most products are screen protectors for iPhone 11 Pro/X/XS (5.8-inch), which is highly relevant to the query - Top items (JETech, Ailun) are particularly relevant as they specifically mention iPhone 11 Pro.
**Position Bias**:- First few positions have higher probability of being selected- Top 3 positions (JETech, JETech, Ailun) have the highest position scores.
**Brand Bias**:- First two positions are JETech brand (high brand bias at top)- This creates a negative shopping experience- Having same brand adjacent to each other reduces purchase likelihood.
**Irrelevance Bias**:- Most items are relevant to the query- However, some products like Trianium (4th position) is for iPhone 11 Pro Max (6.5"), which is incorrect size- Overall irrelevance bias is low as most products match the query.
**Color Bias**:- Many items are listed as "Clear" or "Transparent"- High color similarity among products- This creates a moderate color bias.
**Final Analysis**: Despite high relevance of products, there are several concerns: 1. High brand bias at top positions (JETech repeated)2. High color bias (many transparent/clear options)3. The position bias favors top items, but those have brand repetition 4. Some size mismatches in the list. Given these factors, especially the high brand bias at top positions and color similarity, a frugal user would likely be hesitant to make a purchase from this list.

**D(purchase)** = no; The brand repetition at top positions and lack of diversity in colors significantly harm the shopping experience, making it less likely for a careful shopper to make a purchase, even though the products themselves are relevant to the query.

Figure 7: **LLM response to the initial list of items shown in Figure 6**.

---

**LLM's response to the ranked list provided by REWARDRANK**

Let me analyze this shopping scenario carefully:
1. **Relevance Analysis:**- All products are screen protectors for iPhone 11 Pro/X/XS, which matches the query perfectly- Most products offer multiple packs (2-3 pieces)- All are tempered glass protectors- High relevance across items.
2. **Position Bias**:- Top positions have higher visibility- First three items (UNBREAKcable, TETHYS, Ailun) have strongest position advantage.
3. **Brand Bias**:- JETech appears twice (positions 5 and 6) - creates brand bias- Other brands are well distributed- Moderate brand bias due to JETech repetition.
4. **Irrelevance Bias**:- Low irrelevance bias as all products are highly relevant to query- All products are specifically designed for the requested phone model- Core features (tempered glass, screen protection) are consistent.
5. **Color Bias**:- Most items are clear/transparent- Very similar color schemes across products- High color bias due to lack of variety.
**Final Decision Analysis**:- While relevance is high and position bias favors top items- The high color bias (all transparent) and moderate brand bias (JETech repetition) are concerning- However, given this is a screen protector where transparency is expected/desired, color bias should be weighted less heavily- The first three positions offer different brands and good specifications- TETHYS (2nd position) offers 3-pack with edge-to-edge coverage and installation frame.

**D(purchase)** = yes; Item to be purchased = TETHYS Glass Screen Protector (B07D6XR7FM)Reasoning: Selected TETHYS due to good position (2nd), differentiated brand, comprehensive features, and 3-pack value proposition. The high relevance and acceptable biases make this a reasonable purchase despite some concerns.

Figure 8: **LLM response to the ranked list generated by REWARDRANK**.

Table 7: **Ablation studies for counterfactual evaluation of LTR methods.** This table presents ablation results for RewardRank under different configurations, including variations in SoftSort temperature $\tau$, misspecification correction regularization $\lambda$, and the addition of the auxiliary reward loss term from Eqn 6. We also report results for PG-rank* using different numbers of Monte Carlo samples. The counterfactual evaluation metrics are: $\Pr(\#\text{Clicks} \geq 1)$ for *PO-Eval* and $\Pr(\#\text{Purchase} \geq 1)$ for *LAU-Eval*.

| | PO-Eval | LAU-Eval |
|---|---|---|
| **Method** | $\Pr(\#\text{Clicks} \geq 1)$ | $\Pr(\#\text{Purchase} \geq 1)$ |
| Upper-Bound | 0.553 | - |
| ListNet Cao et al. (2007) | $0.523 \pm 0.0007$ | $0.521 \pm 0.009$ |
| ListMLE Xia et al. (2008) | $0.522 \pm 0.0007$ | $0.522 \pm 0.008$ |
| LambdaRank Wang et al. (2018) | $0.524 \pm 0.0007$ | $0.523 \pm 0.009$ |
| PiRank Swezey et al. (2021) | $0.525 \pm 0.0007$ | $0.528 \pm 0.007$ |
| URCC* | $0.462 \pm 0.0005$ | $0.471 \pm 0.008$ |
| PG-rank* (mc=1) | $0.481 \pm 0.0006$ | $0.441 \pm 0.006$ |
| PG-rank* (mc=5) | $0.495 \pm 0.0005$ | $0.465 \pm 0.007$ |
| PG-rank* (mc=10) | $0.501 \pm 0.0005$ | $0.489 \pm 0.007$ |
| NAR4Rec Ren et al. (2024) | $0.527 \pm 0.0007$ | - |
| GRPO Xu et al. (2022) | $0.518 \pm 0.0005$ | - |
| **SoftSort Temperature $\tau$** | | |
| RewardRank ($\tau = 0.1, \lambda = 0.0$) | $0.531 \pm 0.0005$ | $0.548 \pm 0.008$ |
| RewardRank ($\tau = 0.2, \lambda = 0.0$) | $0.532 \pm 0.0005$ | $0.550 \pm 0.008$ |
| RewardRank ($\tau = 0.5, \lambda = 0.0$) | $0.533 \pm 0.0005$ | $0.551 \pm 0.007$ |
| RewardRank ($\tau = 0.7, \lambda = 0.0$) | $0.531 \pm 0.0005$ | $0.550 \pm 0.008$ |
| RewardRank ($\tau = 1.0, \lambda = 0.0$) | $0.530 \pm 0.0005$ | $0.549 \pm 0.009$ |
| **Misspecification Correction $\lambda$** | | |
| RewardRank ($\tau = 0.5, \lambda = 0.1$) | $0.532 \pm 0.0005$ | $0.549 \pm 0.007$ |
| RewardRank ($\tau = 0.5, \lambda = 0.3$) | $0.534 \pm 0.0007$ | $0.554 \pm 0.007$ |
| RewardRank ($\tau = 0.5, \lambda = 0.7$) | $0.536 \pm 0.0007$ | $0.561 \pm 0.008$ |
| RewardRank ($\tau = 0.5, \lambda = 1.0$) | $0.533 \pm 0.0007$ | $0.553 \pm 0.006$ |
| **Auxiliary Per-Item Regularizer Eqn 6** | | |
| RewardRank (reward loss = Eqn 3) | $0.528 \pm 0.0005$ | $0.553 \pm 0.008$ |
| RewardRank (reward loss = Eqn 3 + Eqn 6) | $0.536 \pm 0.0005$ | $0.561 \pm 0.008$ |
| **Using pretrained ranker: PiRank** | | |
| URCC* | $0.521 \pm 0.0005$ | - |
| PG-rank* | $0.503 \pm 0.0006$ | - |
| RewardRank | $0.538 \pm 0.0005$ | - |

# D  FURTHER ABLATION STUDIES

We use the Baidu-ULTR dataset to study how the performance of REWARDRANK varies with two key hyperparameters: the temperature $\tau$ of the *SoftSort* operator, which controls permutation sharpness, and the regularization strength $\lambda$ for reward misspecification correction introduced in Eqn 13. Varying $\tau \in \{0.1, 0.2, 0.7, 1.0\}$ shows that moderate temperature ($\tau = 0.2 - 0.5$) achieves the best utility and relevance alignment. Too low a temperature leads to unstable gradients due to near-hard permutations, while higher values oversmooth rankings, diluting learning signals. Fixing $\tau = 0.5$, we ablate the correction term with $\lambda \in \{0.0, 0.1, 0.3, 0.7, 1.0\}$. As shown in Table 7 and visualized in Figure 3, moderate correction ($\lambda = 0.5 - 0.7$) yields the best trade-off, by down-weighting unreliable samples without discarding informative ones. This results in higher IPS utility, confirming the benefit of explicitly mitigating reward misspecification.

We explore the impact of incorporating an auxiliary item-level reward loss (Eqn 6) into the training objective of the reward model. As shown in Table 7, adding this auxiliary loss to the list-level cross-entropy objective (Eqn 3) improves expected utility from 0.528 to 0.536. This indicates that learning

to predict the per-item feedback as an auxiliary task enhances the reward model's generalization and improves the downstream utility-optimized ranking.

Table 7 presents the results for the pretrained ranker, which is the ranker trained with PiRank Swezey et al. (2021) LTR loss. URCC$^*$, being dependent on the pretrained ranker, demonstrates larger performance improvements. However, the gains from the pre-trained ranker are not as significant, suggesting that URCC$^*$ 's performance is more sensitive to the quality of the pretrained model. On the other hand, REWARDRANK and PG-rank$^*$ show limited improvements when using the pretrained ranker, as their performance is not heavily reliant on the presence of a strong pretrained model. These methods are more robust in their ranking capabilities and do not exhibit substantial gains from a pretrained ranker.

## E    INFERENCE COST AND LIMITATIONS

**Inference Cost.**    The main inference cost in our work arises from using large language models (LLMs) for ranking and purchase probability estimation. These models require significant computational resources, especially for large datasets and permutations of items. Optimizations like batch processing and multi-GPU use help manage costs, but scalability remains a challenge. Caching frequently accessed queries can further reduce repeated computation costs.

**Limitations.**    While both PO-Eval and LAU-Eval provide valuable insights into ranking quality and user preferences, there are inherent limitations in each approach. These limitations arise from their reliance on specific biases and the quality of input data, which may affect their performance in diverse real-world scenarios. Below, we outline the key limitations of each method:

- **PO-Eval Limitations:** While PO-Eval provides a robust baseline for position-debiasing, it is limited in behavioral scope. It primarily focuses on mitigating position bias without considering other nuanced user preferences, such as brand bias or contextual relevance, which can lead to suboptimal performance in more complex scenarios.
- **LAU-Eval Limitations:** LAU-Eval captures richer heuristics and offers more context-aware ranking, but it depends heavily on the quality and stability of the LLM outputs. Inconsistent or noisy outputs from the LLM can negatively affect the reliability of the evaluation, as the method assumes that the LLM accurately reflects user preferences in all scenarios.

These limitations highlight areas for future improvement, such as incorporating additional user behavior modeling and enhancing the robustness of the LLM outputs.

