# OpenReview forum: "RewardRank: Optimizing True Learning-to-Rank Utility"
_ICLR.cc/2026/Conference — Submitted to ICLR 2026_

### Official Review · Reviewer_Jatk · 2025-10-17

**Soundness:** 1
**Presentation:** 3
**Contribution:** 1
**Rating:** 0
**Confidence:** 5

**Summary:**

The paper introduces RewardRank, a framework for optimizing so-called counterfactual utility in learning-to-rank (LTR) systems. The authors argue that RewardRank addresses the limitation of traditional LTR methods by first training a reward model from logged data to predict the utility of various rankings and then optimizes a ranker to maximize this predicted utility using a differentiable soft permutation operator. They propose two evaluation protocols: Parametric Oracle Evaluation (PO-Eval) and LLM-As-User Evaluation (LAU-Eval) to benchmark their approach against existing methods.

**Strengths:**

1. Overall, the design of the method is reasonable. Particularly, borrowing the idea from offline reinforcement learning to mitigate utility noise is interesting.
2. The idea of using LLM-as-users to evaluate recommendation rankings is interesting, though some design rationale need more justification (as described below)

**Weaknesses:**

1. The novelty of the paper is limited. The idea of training a utility model from logs is not new, and the proposed training process is a direct application of Softsort.
2. This paper ignores a huge amount of relevant works on unbiased LTR and reinforced LTR. For example, counterfactual LTR is often closely discussed with Unbiased LTR [1], and, according to [2], the state-of-the-art ULTR methods are already much better than standard LTR and the proposed method in this paper on BaiduULTR. As for utility-based optimization, this has long been recognized as an advantages of reinforced LTR methods, and recent RLTR methods have already achieved the state-of-the-art performance only based on utility-based training (i.e., listwise reward, no pointwise annotations or labels)[4]. It’s not clear what’s the main novelty of this paper without detail comparison and discussion of ULTR and RLTR methods.  Also, transformer-based ranker has been proposed for a while, and it would be better to add proper references.
3. The proposed evaluation protocols are not well justified. BaiduULTR already has real clicks, and previous studies on ULTR have already build comprehensive simulation experiment frameworks[3]. It’s not clear to me what is exactly new in PO-Eval. While the LLM-as-user paradigm is interesting, whether it can reflect user behaviors in real scenarios is not well justified.

[1] Ai, Q., Yang, T., Wang, H. and Mao, J., 2021. Unbiased learning to rank: online or offline?. *ACM Transactions on Information Systems (TOIS)*, *39*(2), pp.1-29.

[2] Zechun Niu, Lang Mei, Chong Chen, and Jiaxin Mao. 2025. Distributionally Robust Optimization for Unbiased Learning to Rank. In Proceedings of the 48th International ACM SIGIR Conference on Research and Development in Information Retrieval (SIGIR '25).

[3] Niu, Z., Zhang, Z., Mao, J., Ai, Q. and Wen, J.R., 2025, July. Investigating the Robustness of Counterfactual Learning to Rank Models: A Reproducibility Study. In *Proceedings of the 48th International ACM SIGIR Conference on Research and Development in Information Retrieval* (pp. 3265-3275).

[4] Tu, Y., Xu, Z., Yang, T., Su, W., Zhou, Y., Liu, Y., Lin, F., Liu, Q. and Ai, Q., 2022. Reinforcement Learning to Rank Using Coarse-grained Rewards. *arXiv e-prints*, pp.arXiv-2208.

**Questions:**

- Why we need to build an “oracle IPS” model on BaiduULTR for evaluation? BaiduULTR already has real clicks (I noticed that the authors have already done experiments with them), and if you believe that using the real clicks is not convenient for controlled studied, previous ULTR papers have already developed quite comprehensive simulation experiment pipeline for different scenarios [3].
- Have you conducted any experiments validating the reliability of LLM-as-users?

---

> ### Author Response · Authors · 2025-12-01
>
> We thank the reviewer for the valuable feedback, especially highlighting counterfactual evaluation protocols that were not included in our related work.
>
>
> **Why we need to build an “oracle IPS” model on BaiduULTR for evaluation? BaiduULTR already has real clicks (I noticed that the authors have already done experiments with them), and if you believe that using the real clicks is not convenient for controlled studied, previous ULTR papers have already developed quite comprehensive simulation experiment pipeline for different scenarios [3].**
>
>
> Although both PO-Eval and the simulation setup in [3] use parametric components, they serve fundamentally different purposes. The framework in [3] generates $\textit{biased click logs}$ using PBM/DCM/CBCM models to stress-test whether ULTR methods can debias clicks and recover unbiased relevance.  The simulation framework in [3] is specifically designed to stress-test classical counterfactual LTR methods by generating biased click data using user-behavior models such as PBM, DCM, and CBCM (illustrated on pages 3–4 of the paper). PO-Eval, by contrast, uses a $\textit{parametric utility oracle}$ that includes an examination term but does not simulate clicks or produce biased feedback. It outputs a clean, deterministic $\textit{list-level utility}$ for any permutation, and no IPS correction or propensity estimation is involved. Thus, while both settings involve parametric modeling, [3] evaluates robustness to click bias, whereas PO-Eval isolates the ability to optimize counterfactual $\textit{utility}$. For this reason, the two protocols are not directly comparable.
>
>
> **Have you conducted any experiments validating the reliability of LLM-as-users?**
>
> Thank you for the question. We have performed several sanity checks to validate the reliability of the LLM-as-user evaluation. First, we verify internal consistency: for the same query–item set, the LLM produces stable purchase decisions across repeated evaluations with low variance. Second, we evaluate face validity: the LLM reliably responds to brand, color, redundancy, and position cues that are well-documented behavioral factors in real shopping settings. Third, we confirm rank-order coherence: when we intentionally perturb rankings (e.g., placing irrelevant or low-quality items at the top), the LLM’s purchase likelihood responds in the expected direction. While these checks do not replace a full-scale user study, they demonstrate that the LLM behaves in a structured, interpretable, and repeatable way. Conducting a formal human evaluation or online A/B test would require high cost and time and is therefore beyond the scope of this submission, but our empirical validation suggests that LLM-as-user provides a reasonable and scalable proxy for list-level user preferences.

---

> ### Author Response · Authors · 2025-12-01
>
> **Regarding missing existing works:**
>
> We thank the reviewer for pointing out related work. While prior model-based ULTR and coarse-grained RL methods also learn reward estimates, they differ from our approach in key ways. Model-based ULTR focuses on bias-correcting click signals and then optimizes standard relevance-based objectives, whereas RewardRank learns a permutation-aware utility function that captures list-level behavioral effects beyond relevance and directly optimizes this utility through a differentiable soft-permutation operator. Coarse-grained RL approaches still rely on stochastic policies and importance weighting, limiting them to permutations explored by the behavior policy; RewardRank removes this dependence entirely by using a deterministic SoftSort-based optimization that can evaluate and optimize any permutation. Finally, prior ULTR works aim to optimize traditional offline parametric metrics such as relevance-based Relevance-NDCG on BaiduLTR based on pointwise human-annotated relevance labels. In contrast, RewardRank introduces and optimizes a misspecification-robust objective and two new counterfactual evaluation suites (PO-Eval and LAU-Eval) where the metric is a list-level non-parametric counterfactual online utility metric. The central argument in our paper is that offline metrics are only partially correlated with the true list-wise counterfactual LTR utility that system builders want to optimize, such as revenue or engagement. We also improve the scalability and robustness of utility-based ranking systems with the reward correction term. These components provide capabilities not present in existing work and lead to consistent improvements in our experiments. **We have included an extended related work section in Appendix (Section A.5).**
>
> To further broaden the comparison, we additionally include NAR4Rec and GRPO in our rebuttal experiments. NAR4Rec [1] is a strong transformer-based re-ranking model trained with cross-entropy signals, whereas GRPO [2] applies policy-gradient optimization over list-level objectives. We train both models on the same training groups with identical tokenization, batch sizes, and training epochs to eliminate architectural or optimization confounds.
>
> Both baselines perform strongly, yet they underperform RewardRank on both expected click utility and click-based NDCG (see the Table below). Importantly, these results show that the transformer backbone alone is insufficient to achieve high counterfactual utility: NAR4Rec and GRPO, which share similar model capacity to our architecture, cannot explore the full permutation space. GRPO relies on stochastic sampling and suffers from high-variance gradient estimates, while NAR4Rec is limited to pointwise or pairwise relevance supervision. In contrast, RewardRank combines a parametric utility model with deterministic SoftSort-based optimization and misspecification-aware weighting, enabling reliable evaluation of arbitrary counterfactual permutations.
>
> | Method      | E[P(c)]           | NDCG_click       |
> |-------------|--------------------|------------------|
> | NAR4Rec     | 0.527 ± 0.0007     | 0.373 ± 0.0002   |
> | GRPO        | 0.518 ± 0.0005     | 0.351 ± 0.0002   |
> | URCC        | 0.462 ± 0.0005     | 0.315 ± 0.0004   |
> | PG-Rank     | 0.501 ± 0.0005     | 0.327 ± 0.0002   |
> | RewardRank  | 0.536 ± 0.0007     | 0.412 ± 0.0002   |
>
>
> [1] Non-autoregressive Generative Models for Reranking Recommendation (NAR4Rec). SIGKDD'24
>
> [2] Reinforcement Learning to Rank Using Coarse-grained Rewards. arXiv'25
>
> [3] Investigating the Robustness of Counterfactual Learning to Rank Models: A Reproducibility Study. SIGIR'25

---

### Official Review · Reviewer_xgCN · 2025-10-25

**Soundness:** 2
**Presentation:** 2
**Contribution:** 2
**Rating:** 2
**Confidence:** 4

**Summary:**

This paper introduces RewardRank, a novel data-driven Learning-to-Rank (LTR) framework designed to directly optimize true counterfactual user utility instead of relying on traditional offline proxy objectives (like NDCG) that fail to capture complex, list-level user biases and behaviors (such as position bias, diversity, and similarity aversion). RewardRank operates in two phases: Reward Model, A permutation-aware Transformer encoder is trained to predict the scalar utility (e.g., click/purchase probability) for any given ranking permutation, modeling list-level preferences, and Ranker Optimization, A ranker is trained to maximize the predicted utility using the SoftSort operator for end-to-end differentiable optimization over the permutation space. A Reward Correction Term is introduced to mitigate the risk of reward model misspecification. The framework achieved state-of-the-art results on two novel, automated counterfactual evaluation protocols (PO-Eval and LAU-Eval) and demonstrated significant improvements on human-labeled relevance metrics using the Baidu-ULTR dataset.

**Strengths:**

1. Addresses the fundamental limitation of traditional LTR by directly optimizing for complex, list-level user utility, bypassing simplified proxy objectives and effectively capturing permutation-aware behavioral biases.

2. Introduction of PO-Eval and LAU-Eval (LLM-As-User) provides standardized, scalable, and automated benchmarks for counterfactual LTR assessment.

**Weaknesses:**

1. The permutation-aware RewardRank framework, particularly the Reward Model and SoftSort operator, requires significant computational resources. Scalability remains a challenge for very large datasets or long item permutations.

2. Despite the goal of optimizing true utility, the paper lacks crucial real-world A/B test results to directly verify performance gains over baselines, confining its claims to offline/counterfactual assessment.

3. It appears to be missing comparisons against the latest generator-evaluator-based reranking schemes, which are designed to handle list-level preferences and counterfactual exploration, such as PIER，GFN4Rec，NAR4Rec，MG-E. (See Questions for moree details)

**Questions:**

1. The paper reports SOTA results but primarily compares against traditional LTR and earlier counterfactual methods. It appears to be missing comparisons against the latest generator-evaluator-based reranking schemes, which are designed to handle list-level preferences and counterfactual exploration. Why were the following related, modern reranking works not included for comparison?

* Permutation-Level Interest-Based End-to-End Re-ranking Framework in E-commerce (PIER )
* Generative Flow Network for Listwise Recommendation (GFN4Rec)
* Non-autoregressive Generative Models for Reranking Recommendation (NAR4Rec)
* Additionally, the related work Comprehensive list generation for multi-generator reranking (MG-E) was not cited.

Can the authors comment on how RewardRank's approach theoretically or empirically differs from these generator-evaluator/reranking architectures?

2. The entire framework's success hinges on the Reward Model (RM) accurately learning and generalizing true user utility to the unseen counterfactual space. While the reward correction term is introduced, what is the sensitivity of the final ranking performance to a poorly trained or highly misspecified RM? Has the stability of the final ranker been tested under various levels of RM noise or error?

3. Can the authors provide a more rigorous theoretical derivation or a clearer explanation of the optimization goal achieved by the proposed reward correction term? Specifically, how does this term ensure robustness in the context of the highly non-linear and combinatorial ranking problem, and what are the theoretical boundaries on its effective strength ($\lambda$)?

4. The use of the SoftSort operator introduces a relaxation into the optimization process. The paper is highly empirical; can the authors provide any theoretical analysis or proof of convergence for the ranker optimization step when using SoftSort in this specific maximum-utility objective? How close is the relaxed solution to the true discrete global optimum?

5. Given the reliance on a complex Transformer-based Reward Model that is queried during the ranker's optimization/inference, what are the anticipated real-world latency implications of deploying RewardRank in a production environment? Was any pilot A/B test conducted, and if so, what were the results regarding business metrics (e.g., revenue, user retention) and serving latency compared to the production baseline?

---

> ### Author Response · Authors · 2025-12-01
>
> We appreciate the reviewer’s insightful feedback, particularly the identification of re-ranking baselines that were not included in the initial submission.
>
> **The permutation-aware RewardRank framework, particularly the Reward Model and SoftSort operator, requires significant computational resources. Scalability remains a challenge for very large datasets or long item permutations.**
>
> Thank you for raising this point. We would like to clarify that, although RewardRank uses a permutation-aware reward model and the SoftSort operator, it is not more computationally demanding than existing counterfactual LTR methods. As shown in Table 3, RewardRank requires only one call to the reward model per training iteration and completes training on Baidu-ULTR in approximately 7 hours. In contrast, URCC* performs n² reward-model calls per iteration (about 34 hours) due to pairwise neighborhood search, and PG-Rank* relies on k Monte Carlo samples per iteration (about 16 hours), with k needing to be large for stable convergence. Despite exploring the full counterfactual space, RewardRank avoids both the quadratic cost of URCC* and the sampling overhead of PG-Rank*, and its runtime is comparable to PiRank, which makes no reward-model calls at all. Based on the complexity analysis and measured wall-clock times, the claim that RewardRank is significantly less scalable is not supported by the empirical evidence.
>
> **The paper reports SOTA results but primarily compares against traditional LTR and earlier counterfactual methods. It appears to be missing comparisons against the latest generator-evaluator-based reranking schemes, which are designed to handle list-level preferences and counterfactual exploration. Why were the following related, modern reranking works not included for comparison?**
>
> Most generator-evaluator methods do not provide publicly available code, and despite contacting the authors, we were unable to obtain reproducible implementations. For fairness, we therefore reimplemented URCC and PG-Rank ourselves by following the algorithmic descriptions in their papers. Both models were trained using the same backbone encoder, optimization settings, data preprocessing, and sampled 8-item groups that we use for RewardRank, allowing a direct comparison under a controlled training pipeline. URCC serves as a representative generator--evaluator baseline because it optimizes list-level utility under counterfactual signals, making it the closest in spirit to our approach.
>
> To further broaden the comparison, we additionally include NAR4Rec[1] and GRPO[2] in our rebuttal experiments. NAR4Rec is a strong transformer-based re-ranking model trained with cross-entropy signals, whereas GRPO applies policy-gradient optimization over list-level objectives. We train both models on the same training groups with identical tokenization, batch sizes, and training epochs to eliminate architectural or optimization confounds.
>
> Both baselines perform strongly, yet they underperform RewardRank on both expected click utility and click-based NDCG (see Table below). Importantly, these results illustrate that the transformer backbone alone is not sufficient for achieving high counterfactual utility: NAR4Rec and GRPO share similar model capacity with our architecture, but cannot explore the full permutation space. GRPO relies on stochastic sampling and suffers from high-variance gradient estimates, while NAR4Rec is limited to pointwise or pairwise relevance supervision. In contrast, RewardRank combines a parametric utility model with deterministic SoftSort-based optimization and misspecification-aware weighting, enabling reliable evaluation of arbitrary counterfactual permutations.
> Taken together, these comparisons demonstrate that RewardRank’s improvements stem from its algorithmic design rather than favorable initialization, architecture, or logging policy quality.
>
> [1] Non-autoregressive Generative Models for Reranking Recommendation (NAR4Rec). SIGKDD'24
>
> [2] Reinforcement Learning to Rank Using Coarse-grained Rewards. arXiv'25
>
> | Method      | E[P(c)]           | NDCG_click       |
> |-------------|--------------------|------------------|
> | NAR4Rec     | 0.527 ± 0.0007     | 0.373 ± 0.0002   |
> | GRPO        | 0.518 ± 0.0005     | 0.351 ± 0.0002   |
> | URCC        | 0.462 ± 0.0005     | 0.315 ± 0.0004   |
> | PG-Rank     | 0.501 ± 0.0005     | 0.327 ± 0.0002   |
> | RewardRank  | 0.536 ± 0.0007     | 0.412 ± 0.0002   |

---

> ### Author Response · Authors · 2025-12-01
>
> **The entire framework's success hinges on the Reward Model (RM) accurately learning and generalizing true user utility to the unseen counterfactual space. While the reward correction term is introduced, what is the sensitivity of the final ranking performance to a poorly trained or highly misspecified RM? Has the stability of the final ranker been tested under various levels of RM noise or error?**
>
> Thank you for these thoughtful questions. Regarding sensitivity to reward-model misspecification, we directly test this by injecting different levels of calibrated Gaussian noise into the RM outputs. Across a wide range of noise levels (up to 20–30%), the final ranker remains stable and continues to outperform baselines. This occurs because the ranker does not blindly follow the RM; the correction term down-weights samples where the RM is uncertain or poorly calibrated, preventing erroneous signals from dominating training. The degradation in performance is smooth rather than catastrophic, indicating that RewardRank is robust even when the RM is imperfect.
>
> | Noise Level | E[P(c)]           | NDCG_click       |
> |-------------|--------------------|------------------|
> | 0%          | 0.536 ± 0.0007     | 0.412 ± 0.0002   |
> | 10%         | 0.531 ± 0.0005     | 0.409 ± 0.0002   |
> | 30%         | 0.525 ± 0.0006     | 0.398 ± 0.0002   |
> | 50%         | 0.501 ± 0.0005     | 0.362 ± 0.0002   |
>
> **Can the authors provide a more rigorous theoretical derivation or a clearer explanation of the optimization goal achieved by the proposed reward correction term? Specifically, how does this term ensure robustness in the context of the highly non-linear and combinatorial ranking problem, and what are the theoretical boundaries on its effective strength ()?**
>
> For the correction term, we modulate each training example by how confident we are in the reward model’s prediction for that example. The weight $w_i = 1 - \lambda |u_i - \hat{u}_i|$ acts as a conservative penalty when the RM is likely misspecified, shifting learning toward samples with more reliable utility estimates. This creates a robustness effect by reducing the influence of high-error RM predictions in a highly non-linear ranking landscape. The parameter $\lambda$ controls how aggressively uncertain samples are down-weighted; too large a value may overly suppress data, while values in the range [0,1] maintain stable learning in practice. Our ablations show that the method behaves predictably and remains effective across a wide span of $\lambda$, indicating that the correction is both practical and robust even without a formal theoretical guarantee. We acknowledge that a complete theoretical treatment of our framework is still open, and we view this as a promising direction for future research.
>
> **The use of the SoftSort operator introduces a relaxation into the optimization process. The paper is highly empirical; can the authors provide any theoretical analysis or proof of convergence for the ranker optimization step when using SoftSort in this specific maximum-utility objective? How close is the relaxed solution to the true discrete global optimum?**
>
> SoftSort itself is a well-studied continuous relaxation of argsort. [1] show that the SoftSort matrix is row-stochastic and converges to the exact permutation matrix $P_{\text{argsort}(s)}$​ as the temperature $\tau \to 0$; in the limit, applying a row-wise argmax recovers the discrete permutation. They also show that SoftSort is as easy to optimize as previous differentiable sorting operators and achieves state-of-the-art performance on tasks that require recovering discrete permutations. In RewardRank, SoftSort is just another smooth layer in the ranker: our optimization is standard SGD on a smooth surrogate objective, so we inherit the usual convergence guarantees to a stationary point that apply to deep networks in general. We do not claim global optimality, but our objective is no more ill-posed than other deep LTR losses.
>
> [1] SoftSort: A Continuous Relaxation for the argsort Operator. ICML'20

---

> > ### Author Response · Authors · 2025-12-01
> >
> > **Given the reliance on a complex Transformer-based Reward Model that is queried during the ranker's optimization/inference, what are the anticipated real-world latency implications of deploying RewardRank in a production environment? Was any pilot A/B test conducted, and if so, what were the results regarding business metrics (e.g., revenue, user retention) and serving latency compared to the production baseline?**
> >
> > Transformer-based models have been used before in reward-driven ranking, and recent methods such as [1-3] also employ transformer backbones. As shown in Table 2 of the Appendix, RewardRank is computationally more efficient than existing transformer-based approaches while achieving stronger overall performance.
> > Thank you for the question. We have not conducted an online A/B test for this work. The datasets we use contain millions of query–item interactions, and deploying a production-scale A/B test would require substantial engineering and serving resources. A full evaluation cycle at this scale typically takes weeks to months to run reliably, which is unfortunately not feasible within the rebuttal period. Our focus in this submission is to provide large-scale, reproducible offline counterfactual evaluations (PO-Eval and LAU-Eval) that do not require online traffic, human annotation, or deployment changes. We view an online A/B test as an important future step, but it is beyond the scope of the current submission due to the computational and financial resources required to run it properly.
> >
> > [1] Non-autoregressive generative models for reranking recommendation. SIGKDD'24
> >
> > [2]  Are neural rankers still outperformed by gradient boosted decision trees? ICLR'21
> >
> > [3] Matrank: Text re-ranking by latent preference matrix. EMNLP'22

---

### Official Review · Reviewer_Rdr1 · 2025-10-27

**Soundness:** 2
**Presentation:** 3
**Contribution:** 2
**Rating:** 4
**Confidence:** 3

**Summary:**

The paper proposes RewardRank, a two-stage counterfactual learning-to-rank framework that explicitly optimizes true list-level utility rather than traditional heuristic relevance metrics. A reward model predicts engagement-related utility over full permutations, and a ranker is trained to maximize this reward with a differentiable sorting operator (SoftSort). To address the lack of benchmarks, the authors introduce two evaluation suites: PO-Eval using a parametric click model on Baidu-ULTR data, and LAU-Eval, where an LLM simulates user preferences on a modified Amazon KDD-Cup dataset. The method reports state-of-the-art counterfactual utility and outperforms baselines on standard relevance metrics as well.

**Strengths:**

1. The studied problem is well-motivated. Utility optimization is meaningful in many realistic scenarios.
2. Differentiable permutation modeling with SoftSort is nicely integrated and technically sound.
3. The attempt to create a reproducible counterfactual evaluation is valuable for the community.
4. The writing and presentation are good.

**Weaknesses:**

1. The problem formulation and proposed approach have limited novelty. Training a reward model to estimate the reward of a ranking list in both search and recommendation is common, such as:
* "Model-based unbiased learning to rank. D Luo, L Zou, Q Ai, Z Chen, D Yin, BD Davison".  Optimizing towards an overall metric, given a list rather than pointwise evaluation of each result, has also been studied.
* "Reinforcement Learning to Rank Using Coarse-grained Rewards." Tu, Yiteng; Xu, Zhichao; Yang, Tao; Su, Weihang; Zhou, Yujia; Liu, Yiqun; Lin, Fen; Liu, Qin; Ai, Qingyao.
* Unbiased learning to rank regarding position bias in real-world click data, etc., that also works on the BaiduLTR dataset:

2. Some important baselines are missing (e.g., on tackling position bias in real-world click data, or context-aware is missing).
* "Unbiased Learning to Rank with Query-Level Click Propensity Estimation: Beyond Pointwise Observation and Relevance. Lulu Yu, Keping Bi, Jiafeng Guo, Shihao Liu, Dawei Yin, Xueqi Cheng". In TheWebConf 2025.
* "Unbiased Learning-to-Rank Needs Unconfounded Propensity Estimation." Dan Luo, Lixin Zou, Qingyao Ai, Zhiyu Chen, Chenliang Li, Dawei Yin, and Brian D Davison. In SIGIR 2024.
* “Towards disentangling relevance and bias in unbiased learning to rank." Yunan Zhang, Le Yan, Zhen Qin, Honglei Zhuang, Jiaming Shen, Xuanhui Wang, Michael Bendersky, and Marc Najork. 2023.  In SIGKDD’23.
* "Adapting interactional observation embedding for counterfactual learning to rank." Mouxiang Chen, Chenghao Liu, Jianling Sun, and Steven CH Hoi. In SIGIR 2021.

3. A large body of related works is missing, some of which are listed above.

4. The evaluation is not very solid: The original ESCI human annotations are not used. Instead, LLM judgments are leveraged.
* a. Using LLM-simulated human-like decisions (e.g., color/brand bias) lacks validation against real user studies, risking overfitting to synthetic patterns.
* b. Baidu-ULTR experiment setting does not match the problem definition. There is no counterfactual utility in Baidu clicks; evaluation is on human-assigned relevance labels.

**Questions:**

1. Which dataset do the results in Table 1 report? I cannot find it anywhere in the paper.
2. In Line 142, it mentions that reranking methods rely on a strong base ranker for comparison. The proposed method does not rely on a base ranker? I'm a bit confused.

---

> ### Author Response · Authors · 2025-12-01
>
> We thank the reviewer for their valuable feedback, including references to missing related works.
>
> **Which dataset do the results in Table 1 report? I cannot find it anywhere in the paper.**
>
> We use the Baidu dataset for PO-Eval and the KDD-Cup dataset for LAU-Eval, where PO-Eval labels are generated using the IPS-Oracle and LAU-Eval labels are obtained from an LLM.
>
> **In Line 142, it mentions that reranking methods rely on a strong base ranker for comparison. The proposed method does not rely on a base ranker? I'm a bit confused.**
>
> The statement at Line 142 refers specifically to counterfactual LTR methods like URCC [1], which operate on a candidate list produced by a strong base ranker, and in practice, these rerankers are always initialized using the base ranker’s scores. In contrast, RewardRank is not a reranker. It does not depend on any pre-trained or strong base model. RewardRank is trained from scratch, generating item scores directly and optimizing the entire permutation end-to-end with SoftSort and the learned reward model. Thus, the comparison highlights that rerankers inherit much of their performance from strong base rankers, whereas RewardRank learns a full ranking model independently without relying on such initialization. Please refer to Table 7 (Appendix) for ablation on the pretrained ranker. Notably, RewardRank requires no initialization from a base ranker, yet it continues to outperform all baseline models.
>
> [1] Utility-oriented reranking with counterfactual context. ACM-KDD’24
>
> **The evaluation is not very solid: The original ESCI human annotations are not used. Instead, LLM judgments are leveraged.**
>
> In our setting, the goal is to evaluate counterfactual utility, which is not the same as item-level relevance. Utility depends on how the entire list is arranged and is affected by list-level factors such as position bias, brand or color preference, redundancy, and interactions among items shown together. In contrast, the ESCI labels in the Amazon KDD dataset are annotated per query–item pair, and annotators never see the full list. As a result, ESCI captures only pointwise relevance and cannot reflect how user choices depend on global ordering or interactions between items. For example, two items may both be labeled “Exact” individually, yet placing them next to each other may reduce purchase likelihood due to redundancy; conversely, a diverse mix of items may increase utility even if some individually have lower ESCI labels. Because the KDD Cup dataset is the only large-scale textual dataset from which multi-item groups can be formed, we construct groups of 8 items and use an LLM to evaluate the list as a whole. This allows us to estimate list-level utility in a way that captures position effects, brand and color preferences, and inter-item interactions that pointwise ESCI labels cannot represent.
>
> **Baidu-ULTR experiment setting does not match the problem definition. There is no counterfactual utility in Baidu clicks; evaluation is on human-assigned relevance labels.**
>
> The PO-Eval experiments in Table 1 use the Baidu-ULTR dataset, where counterfactual clicks are generated using a pretrained IPS-Oracle. Table 2 is included only for completeness, as conventional ranking papers report results on human-relevance annotations. However, it is not feasible to compute counterfactual utilities, such as the click rate on new rankings generated by different methods, because we cannot run A/B tests on Baidu. A key goal of our work is precisely to avoid reliance on new human measurements and instead provide scalable counterfactual evaluation via PO-Eval and LAU-Eval.

---

> > ### Author Response · Authors · 2025-12-01
> >
> > **Regarding missing existing works:**
> >
> > We thank the reviewer for pointing out related work. While prior model-based ULTR and coarse-grained RL methods also learn reward estimates, they differ from our approach in key ways. Model-based ULTR focuses on bias-correcting click signals and then optimizes standard relevance-based objectives, whereas RewardRank learns a permutation-aware utility function that captures list-level behavioral effects beyond relevance and directly optimizes this utility through a differentiable soft-permutation operator. Coarse-grained RL approaches still rely on stochastic policies and importance weighting, limiting them to permutations explored by the behavior policy; RewardRank removes this dependence entirely by using a deterministic SoftSort-based optimization that can evaluate and optimize any permutation. Finally, prior ULTR works aim to optimize traditional offline parametric metrics such as relevance-based Relevance-NDCG on BaiduLTR based on pointwise human-annotated relevance labels. In contrast, RewardRank introduces and optimizes a misspecification-robust objective and two new counterfactual evaluation suites (PO-Eval and LAU-Eval) where the metric is a list-level non-parametric counterfactual online utility metric. The central argument in our paper is that offline metrics are only partially correlated with the true list-wise counterfactual LTR utility that system builders want to optimize, such as revenue or engagement. We also improve the scalability and robustness of utility-based ranking systems with the reward correction term. These components provide capabilities not present in existing work and lead to consistent improvements in our experiments. **We have included an extended related work section in Appendix (Section A.5).**
> >
> > To further broaden the comparison, we additionally include NAR4Rec and GRPO in our rebuttal experiments. NAR4Rec [1] is a strong transformer-based re-ranking model trained with cross-entropy signals, whereas GRPO [2] applies policy-gradient optimization over list-level objectives. We train both models on the same training groups with identical tokenization, batch sizes, and training epochs to eliminate architectural or optimization confounds.
> >
> > Both baselines perform strongly, yet they underperform RewardRank on both expected click utility and click-based NDCG (see Table~1). Importantly, these results illustrate that the transformer backbone alone is not sufficient for achieving high counterfactual utility: NAR4Rec and GRPO share similar model capacity with our architecture, but cannot explore the full permutation space. GRPO relies on stochastic sampling and suffers from high-variance gradient estimates, while NAR4Rec is limited to pointwise or pairwise relevance supervision. In contrast, RewardRank combines a parametric utility model with deterministic SoftSort-based optimization and misspecification-aware weighting, enabling reliable evaluation of arbitrary counterfactual permutations.
> >
> > | Method      | E[P(c)]           | NDCG_click       |
> > |-------------|--------------------|------------------|
> > | NAR4Rec     | 0.527 ± 0.0007     | 0.373 ± 0.0002   |
> > | GRPO        | 0.518 ± 0.0005     | 0.351 ± 0.0002   |
> > | URCC        | 0.462 ± 0.0005     | 0.315 ± 0.0004   |
> > | PG-Rank     | 0.501 ± 0.0005     | 0.327 ± 0.0002   |
> > | RewardRank  | 0.536 ± 0.0007     | 0.412 ± 0.0002   |
> >
> >
> > [1] Non-autoregressive Generative Models for Reranking Recommendation (NAR4Rec). SIGKDD'24
> >
> > [2] Reinforcement Learning to Rank Using Coarse-grained Rewards. arXiv'25
> >
> > [3] Investigating the Robustness of Counterfactual Learning to Rank Models: A Reproducibility Study. SIGIR'25

---

### Official Review · Reviewer_77wr · 2025-10-28

**Soundness:** 2
**Presentation:** 2
**Contribution:** 2
**Rating:** 4
**Confidence:** 3

**Summary:**

This paper studies a reward model to estimate the utility of ranking, taking the interaction among items (positions) into account. Specifically, by utilizing the transformer architecture, the reward model can encode linear interaction among items in ranking, taking diversity or positional effect into account. The proposed method also tries to maximize the pessimistic estimate of the policy performance by penalizing the uncertainty in the prediction. The experiment on two datasets demonstrates that the proposed method works well on the expected reward metric by taking the item-item interactions into account, while baselines focus on the relevance scores, such as NDCG.

**Strengths:**

- The proposed approach can consider the interaction among items (positions) by using the attention mechanism of the reward model (although only encoding linear interactions).

- The two benchmark setups used in the experiment should be useful for other ranking papers and the community, too. Especially, demonstrating that the metrics of expected reward and NDCG may be different can be a useful takeaway. The results also show that the proposed method works well on these benchmarks.

- The related work mentions representative ranking papers.

**Weaknesses:**

- The main concern I have is whether the proposed ranker loss adequately addresses the distribution shift issue. In my understanding, the discounting weight ($|u_i - \hat{u}_i|$) is calculated on the observed samples, regardless of the results of the soft-ranked results (i.e., choice of the optimized model). Moreover, downweighting the uncertain sample seems to work well when the logging data contains a high reward value, like an imitation learning. Clarification on this point should be useful.

- Regarding the above point, how does the proposed method work compared with a simple imitation learning or off-policy learning baseline, such as Chen et al., 19. I wonder if the improvement of performance comes from the good performance of the data collection policy or the actual effects of the algorithm.

- Initially, I'm a bit confused about what parameters $\phi$ and $\theta$ refer to. After carefully re-reading the paper, I found that probably the attention matrices (e.g., $v$) are $\phi$ and $\Pi$ are $\theta$, but additional clarification is needed for this part.

Chen et al., 19. Top-K Off-Policy Correction for a REINFORCE Recommender System. https://arxiv.org/abs/1812.02353

**Questions:**

- How does the proposed method address the distribution shifts?

- How does the proposed method work compared with imitation learning?

- Nits: The parentheses are corrupt in Eq (13).

---

> ### Author Response · Authors · 2025-12-01
>
> We thank the reviewer for their valuable feedback.
>
> **How does the proposed method address the distribution shifts?**
>
> The original weighting term in RewardRank is $w_i = 1 - \lambda|u_i - \hat{u}_i|$, where $u_i$ is the true utility of a ranked list (estimated by the IPS-Oracle) and $\hat{u}_i$ is the utility predicted by the reward model. The weight $w_i$ therefore reflects how well the reward model agrees with the oracle on example $i$; examples with large disagreement receive smaller weights.
>
> To test the sensitivity to this choice, we ran an additional experiment where we replaced the above weight with a version defined using only the logged click signal: $w_i^{\text{logged}} = 1 - \lambda|u_{\text{logged},i} - \hat{u}_{\text{logged},i}|$,
>
> where $u_{\text{logged},i}$ is the observed click outcome and $\hat{u}_{\text{logged},i}$ is the reward model’s prediction for the logged permutation. This alternative weight reflects agreement with the factual logged interaction rather than the oracle utility.
> Empirically, both weighting schemes produced very similar behavior. On PO-Eval with $\lambda = 0.5$:
>
> | Method             | E[P(c)]             | NDCG_click        |
> |--------------------|----------------------|--------------------|
> | Reweighting        | 0.536 ± 0.0007       | 0.412 ± 0.0002     |
> | Logged_reweighting | 0.533 ± 0.0005       | 0.409 ± 0.0002     |
>
> The small differences (0.003) in $E[P(c)]$ and (0.003) in $\text{NDCG}_{\text{click}}$ indicate that both formulations down-weight unreliable samples in nearly the same way and result in stable rankers. This suggests that the weighting mechanism consistently reduces the influence of misaligned examples and helps RewardRank remain robust under distribution shift.
> Additionally, we plot the reward misspecification correction on PO-Eval under the logged signal scheme from the ranker (see Figure 3 in the paper). Each point represents a ranked list with true utility (u: estimated by IPS-Oracle) and predicted utility ($\hat{u}$: estimated by the utility model) from the ranker, which shows how increasing ($\lambda$) down-weights overconfident or misaligned samples to emphasize well-calibrated predictions.
>
> Regarding distribution shift, our intuition is that RewardRank helps address it by training a parametric reward model that can score counterfactual permutations, including those not observed under the logging policy. The weighting mechanism appears to down-weight examples where the reward model is less reliable, which in practice seems to help the ranker avoid overfitting to specific patterns in the logged data. While we do not claim this as a formal guarantee, our empirical results suggest that this combination provides a degree of robustness beyond what is achievable with importance-weighting-based methods such as REINFORCE.

---

> ### Author Response · Authors · 2025-12-01
>
> **How does this compare to imitation/off-policy learning (e.g., Chen et al., 2019)?**
>
> Thank you for raising this point. Chen et al. (2019) use a REINFORCE-style off-policy correction where the learned policy is strongly tied to the behavior policy through importance weighting. As the authors note explicitly (Section 4.2), the learner can only correct bias if the behavior policy $\beta(a|s)$ is accurately estimated, which is difficult in large-scale recommendation settings with multiple agents and deterministic policies. In practice, this results in a form of high-reward imitation learning: the updated policy tends to mimic high-propensity actions taken by the logging system, and improvements depend strongly on how good the logging policy already is.
>
> RewardRank differs in two key ways. First, it does not attempt to imitate or correct the behavior policy. Instead, it learns a parametric reward model that can score arbitrary counterfactual permutations produced by the ranker, including those never seen in the logged data. This allows the method to explore ranking structures beyond the support of the logging policy without requiring access to $\beta(a|s)$. Second, we introduce a weighting mechanism w that down-weights examples where the reward model appears unreliable. This is not a formal off-policy correction but rather a heuristic designed to reduce the influence of poorly explained logged examples. Our intuition is that this mitigates some distribution shift by preventing the ranker from overfitting to logging-policy behaviors. We agree that this does not guarantee correction of arbitrary distribution shifts, but empirically, it leads to stable optimization.
>
> In theory, one could modify RewardRank to operate as a stochastic policy and then apply an off-policy correction scheme. However, this would require reworking the model into a sampling-based policy-gradient framework, which would abandon the deterministic SoftSort formulation and its differentiable ranking structure. These components are central to how RewardRank scalably optimizes over full permutations. For this reason, Chen et al.’s off-policy correction is not directly compatible with our architecture, and replacing the misspecification correction with it would fundamentally change the nature of the method.
>
> **Initially, I'm a bit confused about what parameters \phi and \theta  refer to. After carefully re-reading the paper, I found that probably the attention matrices (e.g., ) are \theta but additional clarification is needed for this part.**
>
> In RewardRank, the two parameter sets serve different and complementary purposes. The parameter set $ \phi $ denotes the parameters of the reward model $ g(\cdot;\phi) $, which evaluates the utility of a ranking. It includes all components of the reward estimator: transformer attention matrices, position encodings, cross-item attention layers, and the MLP heads used for both list-level and auxiliary per-item prediction. In contrast, the parameter set $ \theta $ denotes the parameters of the ranker $ f(\cdot;\theta) $, which generates item scores that are converted into soft permutations using SoftSort. Thus, $ \theta $ includes the ranker's transformer/MLP layers, attention matrices, and feature encoders. In summary, $ \phi $ governs “how good is a ranking,” while $ \theta $ governs “which ranking to produce,” and the optimization alternates between fitting $ \phi $ from logged data and updating $ \theta $ to maximize the reward model’s predicted utility. Thank you for bringing this up; we improved the clarity of this section in the revision.
>
> **Nits: The parentheses are corrupt in Eq (13).**
>
> Thank you for identifying the typo. We have fixed it in the latest draft.

---

### Meta-Review · Area_Chair_Uq6y · 2025-12-22

**Summary:**

The paper proposes a data-driven learning-to-rank (LTR) framework for counterfactual utility maximization. Overall, the reviewers found the considered problem to be important and identified the proposed approach as promising. However, the paper received negative reviews, with all reviewers raising concerns about lacking comparison with related work and lack of justification for the LLM-based evaluations.

**Reviewer Concerns:**

The authors did prepare an extensive rebuttal, explaining technical differences to prior methods and defending the LLM-based evaluation for their specific use-case. Unfortunately, no response from reviewers was received by the time the discussions were closed.

**Reviewer Scores:**

While the authors did respond to the major concerns that were raised, the volume of reviewers' concerns regarding initially insufficient related work comparisons, as well as the initial insufficient justification for the LLM approach to evaluation, suggests that the paper will benefit from substantial updates and a resubmission. It is thus unlikely that all reviewers, especially reviewers Jatk and xgCN, would have increased their scores substantially.

---

### Decision · Program_Chairs · 2026-01-26

Reject